# IF1 is a cold-regulated switch of ATP synthase hydrolytic activity to support thermogenesis in brown fat

Henver S Brunetta[1,2], Anna S Jung [2,3], Fernando Valdivieso-Rivera [4], Stepheny C de Campos Zani[1], Joel Guerra[2,3], Vanessa O Furino[4], Annelise Francisco[5], Marcelo Berçot[6], Pedro M Moraes-Vieira [6,7], Susanne Keipert [8], Martin Jastroch [8], Laurent O Martinez [9], Carlos H Sponton[4], Roger F Castilho [5], Marcelo A Mori [1,4,7 ✉] & Alexander Bartelt [2,3,10,11 ✉]

## Abstract

**While mechanisms controlling uncoupling protein-1 (UCP1) in thermogenic adipocytes play a pivotal role in non-shivering thermogenesis, it remains unclear whether F₁Fo-ATP synthase function is also regulated in brown adipose tissue (BAT). Here, we show that inhibitory factor 1 (IF1, encoded by *Atp5if1*), an inhibitor of ATP synthase hydrolytic activity, is a critical negative regulator of brown adipocyte energy metabolism. In vivo, IF1 levels are diminished in BAT of cold-adapted mice compared to controls. Additionally, the capacity of ATP synthase to generate mitochondrial membrane potential (MMP) through ATP hydrolysis (the so-called "reverse mode" of ATP synthase) is increased in brown fat. In cultured brown adipocytes, IF1 overexpression results in an inability of mitochondria to sustain the MMP upon adrenergic stimulation, leading to a quiescent-like phenotype in brown adipocytes. In mice, adeno-associated virus-mediated IF1 overexpression in BAT suppresses adrenergic-stimulated thermogenesis and decreases mitochondrial respiration in BAT. Taken together, our work identifies downregulation of IF1 upon cold as a critical event for the facilitation of the reverse mode of ATP synthase as well as to enable energetic adaptation of BAT to effectively support non-shivering thermogenesis.**

**Keywords** Adipocytes; Thermogenesis; UCP1; Metabolism; Mitochondria
**Subject Category** Metabolism

## Introduction

One of the primary functions of brown adipose tissue (BAT) is to generate heat through a process called non-shivering thermogenesis (NST) (Cannon and Nedergaard, 2004), which relies on the activation of uncoupling protein-1 (UCP1), a mitochondrial carrier protein that uncouples mitochondrial respiration from ATP synthesis. Activity of UCP1 is inhibited by purine nucleotides and stimulated by fatty acids, resulting in an increase in proton conductance across the inner mitochondrial membrane (Fedorenko et al, 2012; Fromme et al, 2018; Brunetta et al, 2020; Nicholls, 2021). Thus, activation of UCP1 lowers the mitochondrial membrane potential (MMP), increases the activity of the electron transport chain, and enhances mitochondrial oxygen consumption. As a result, cold exposure and adrenergic activation of BAT lead to increased whole-body oxygen consumption and energy expenditure (Bartelt et al, 2011; Politis-Barber et al, 2022).

Although the regulation of UCP1 is relatively well understood (Matthias et al, 1999; Cannon and Nedergaard, 2004; Nicholls, 2021; Chouchani et al, 2019; Nicholls, 2023), less is known regarding the mechanisms controlling F₁Fₒ-ATP synthase (hereafter called ATP synthase) activity during NST. According to the chemiosmotic theory, the protonmotive force (PMF; proton electrochemical gradient) generated by the activity of the electron transport chain is coupled to ATP production by ATP synthase (Mitchell, 1961). ATP synthase is a rotary motor protein catalyzing ATP synthesis from ADP and inorganic phosphate by using the PMF across the inner mitochondrial membrane. Under specific conditions, ATP synthase also functions as an ATPase (the so-called "reverse mode"), resulting in the formation of ADP plus phosphate and leading to proton transport from the mitochondrial matrix into the intermembrane space (Kobayashi et al, 2023), thus restoring the PMF. Therefore, when PMF is high, Fₒ forcibly rotates

[1]Department of Biochemistry and Tissue Biology, University of Campinas, Campinas, Brazil. [2]Institute for Cardiovascular Prevention (IPEK), Faculty of Medicine, Ludwig-Maximilians-Universität München, Munich, Germany. [3]Institute for Diabetes and Cancer (IDC), Helmholtz Center Munich, Neuherberg, Germany. [4]Obesity and Comorbidities Research Center (OCRC), University of Campinas, Campinas, SP, Brazil. [5]Department of Pathology, University of Campinas, Campinas, SP, Brazil. [6]Department of Genetics, Evolution, Microbiology and Immunology, University of Campinas, Campinas, SP, Brazil. [7]Experimental Medicine Research Cluster (EMRC), University of Campinas, Campinas, SP, Brazil. [8]Department of Molecular Biosciences, The Wenner-Gren Institute, Stockholm University, Stockholm, Sweden. [9]LiMitAging Team, Institute of Metabolic and Cardiovascular Diseases, I2MC UMR1297, IHU HealthAge, INSERM, University of Toulouse, Université Toulouse III - Paul Sabatier (UPS), Toulouse, France. [10]German Center for Cardiovascular Research, Partner Site Munich Heart Alliance, Munich, Germany. [11]German Center for Diabetes Research, Neuherberg, Germany. ✉E-mail: morima@unicamp.br; alexander.bartelt@med.uni-muenchen.de

$F_1$, resulting in ATP synthesis. Conversely, when PMF is low, $F_1$ reverses the rotation and hydrolyzes ATP. Given the vectorial dependence of ATP synthase on the PMF, it is intriguing to investigate how ATP synthase would adapt following modulation of the electrical component of PMF (i.e., the mitochondrial membrane potential) upon UCP1 activation in brown adipocytes.

Located in the mitochondrial matrix, ATP synthase inhibitory factor 1 (encoded by *Atp5if1*, hereafter called IF1), is activated when mitochondrial matrix pH is low, resulting in the inhibition of ATP synthase hydrolytic activity by operating in the reverse mode (Carroll et al, 2024; Cabezón et al, 2003; Gledhill et al, 2007; Pullman and Monroy, 1963; Esparza-Moltó et al, 2017). This mechanism supposedly prevents cellular ATP depletion by mitochondria. However, it is now recognized that under certain conditions, such as low MMP or mitochondrial dysfunction, the reverse mode of ATP synthase is potentiated, generating MMP at the cost of mitochondrial ATP consumption (Acin-Perez et al, 2023; Chen et al, 2014; Nelson et al, 2021). Therefore, the role of IF1 controlling ATP synthase function appears to be more relevant to regulating cellular energy metabolism than previously anticipated (Chen et al, 2014; Formentini et al, 2017; Sánchez-González et al, 2020; Zhou et al, 2022). However, it remains to be determined if IF1 plays a role in BAT energy metabolism during NST.

Here, we investigated the role of IF1 in BAT thermogenic capacity and metabolism by applying in vitro and in vivo gain and loss-of-function experiments in brown adipocytes and mice. In summary, we establish IF1 downregulation as a key adaptive mechanism to modulate brown adipocyte energy metabolism during NST.

# Results

## Cold exposure potentiates the reverse mode of ATP synthase in brown fat

While the adaptive regulation of mitochondrial uncoupling in BAT upon cold exposure is well studied, it remains unclear whether ATP synthase is regulated to support NST. To test this, we first evaluated the hydrolytic activity of ATP synthase operating in reverse mode in BAT from mice exposed to 4 °C for 5 days. As expected, cold-exposed mice lost body mass despite greater food intake compared to mice kept at room temperature (RT, 22 °C) (Fig. EV1A,B). We observed that ATP hydrolysis was low in BAT from animals kept at RT; however, cold exposure increased it by almost 2-fold (Fig. 1A,B). In the absence of ATP or $Mg^{2+}$, or in the presence of oligomycin, a classical inhibitor of $F_oF_1$-ATP synthase, ATP hydrolysis was negligible, confirming the specificity of the assay to measure the hydrolytic activity of ATP synthase (Fig. 1A). In intact mitochondria, the primary consequence of ATP hydrolysis by ATP synthase is pumping protons from the mitochondrial matrix into the intermembrane space. Therefore, we tested the impact of the reverse mode of ATP synthase on MMP in isolated mitochondria from BAT. For that, we isolated mitochondria from the BAT of mice kept at RT or exposed to cold (Fig. 1C) and measured MMP using the fluorescent probe safranin (Fig. 1D). We added antimycin A (to inhibit Complex III) and GDP (to inhibit UCP1) to minimize proton movement across the inner mitochondrial membrane from sources other than ATP synthase activity

itself. Then, we added ATP in the medium to drive ATP synthase into the reverse mode, resulting in the generation of ADP and phosphate and the pumping of protons from the mitochondrial matrix into the intermembrane space. As expected, ATP addition caused an increase in MMP (Fig. 1D), and this occurred in an oligomycin-sensitive manner. However, this increase was larger in mitochondria isolated from BAT of cold-exposed mice (Fig. 1E). In summary, this set of data shows that cold exposure increases the capacity of ATP synthase to function in the reverse mode in BAT.

## Cold exposure lowers IF1 levels in BAT

IF1 is known to inhibit the hydrolytic activity of ATP synthase in conditions when MMP is low or when the mitochondrial matrix becomes too acidic (Cabezon et al, 2000). Therefore, we hypothesized that IF1 could be involved in the regulation of ATP synthase in BAT following cold exposure. When analyzing three different tissues rich in mitochondria (i.e., BAT, liver, and heart), we observed that BAT presents IF1 relative levels comparable with the heart (Fig. 1F,G). We then examined *Atp5if1* mRNA levels in BAT of a different cohort of animals acutely (4 h) exposed to cold or $\beta_3$-adrenergic agonist CL316,243 and found that *Atp5if1* mRNA abundance was lower by approximately 50% following either stimulus (Fig. 1H). Accordingly, we observed that IF1 protein amount was downregulated by ~50% and ~80% following 3 and 5 days of cold exposure, respectively (Fig. 1I,J). Importantly, ATP synthase subunit 5A levels were not changed by cold exposure (Fig. 1I), which suggests that changes in IF1/ATP synthase ratio upon cold exposure (Fig. 1K) are primarily determined by the downregulation of IF1. Of note, cold exposure did not affect IF1 levels in liver and heart (Fig. EV1), indicating a BAT-specific mechanism. In addition, given the marked remodeling mitochondria undergo in BAT following changes in ambient temperature, we also tested IF1 regulation when mice are transferred from RT (22 °C) to thermoneutrality (28 °C) exposure. While UCP1 and complex I subunit NDUFB8 were markedly lower following thermoneutrality adaptation for 3 and 7 days, IF1 levels remained stable (Fig. 1L), suggesting modulation of IF1 levels is specific to BAT adaptation to cold when mice heavily rely on NST (i.e., 4 °C). Together, these findings suggest that the reduction of IF1 protein levels after cold exposure is linked to the greater hydrolytic activity by ATP synthase found in BAT mitochondria from cold-adapted mice (Fig. 1M).

## High IF1 levels lead to collapse of MMP upon adrenergic stimulation

To investigate any causal relationship between IF1 and MMP due to changes in ATP synthase hydrolytic activity, we employed an in vitro IF1 loss-of-function model, in which we silenced *Atp5if1* mRNA levels using small interfering RNA (siRNA) in cultured differentiated WT1 mouse brown adipocytes. First, we validated our experimental approach by using oligomycin and FCCP, both drugs capable of modulating MMP, to verify the accumulation of TMRM within mitochondria in an MMP-dependent manner (Fig. EV2A). Short (i.e., 30 min) pre-treatment with oligomycin increased TMRM accumulation roughly by 20% whereas FCCP decreased it by almost 60%, hence evidencing expected changes in MMP. Interestingly, oligomycin effects over MMP were abrogated

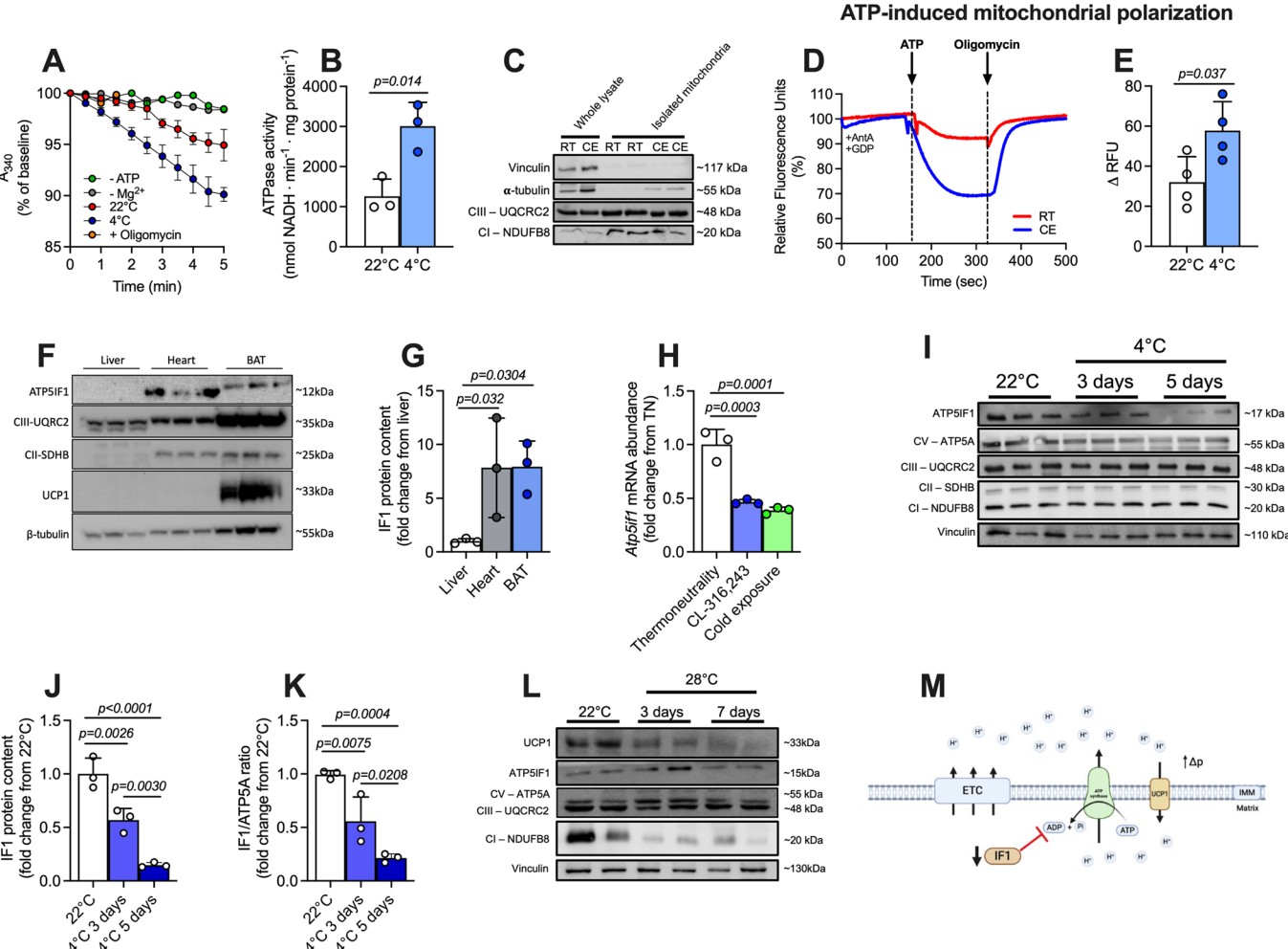

**Figure 1. Cold favors the reverse mode of ATP synthase and lowers IF1 levels in BAT.**

(A) Representative traces of NADH consumption to determine the hydrolytic capacity of ATP synthase and (B) quantification of ATPase activity in BAT from room temperature (RT, 22 °C) or cold-exposed (4 °C) mice (A, B—three biological replicates). (C) Representative immunoblots showing low content of cytosolic proteins in our mitochondrial isolation protocol (two biological replicates). (D) Representative plots of safranin fluorescence in mitochondria isolated from RT or cold-exposed mice in response to ATP addition and (E) quantification of the ATP-induced change in membrane potential (four biological replicates). (F) Representative images and (G) quantification of IF1 protein levels in liver, heart, and BAT of mice kept at RT (three biological replicates). (H) *Atp5if1* mRNA levels in BAT of thermoneutrality-adapted mice, 4 h after cold exposure or CL316,243 injection (three biological replicates). (I) Representative images, (J) quantification of IF1 in BAT following 3 or 5 days of cold exposure and (K) IF1/ATP5A ratio in BAT of animals kept in RT, and 3 or 5 days of cold exposure (three biological replicates). (L) Representative blots of mitochondrial proteins from animals kept at RT and exposed to thermoneutrality (TN) for 3 and 7 days (two biological replicates). (M) Schematic model of the hypothetical relationship between IF1 and UCP1 in brown adipocytes (created with Biorender). A$_{340}$ 340 nm absorbance, ANT adenine nucleotide transporter, ATP adenosine triphosphate, ADP adenosine diphosphate, Mg$^{2+}$ magnesium, AA Antimycin A, CE cold exposure, ATP5IF1 ATP synthase inhibitory factor subunit 1, BAT brown adipose tissue, CIII complex III, CI complex I, ETC electron transport chain, GDP guanosine diphosphate, IF1 inhibitory factor 1, NADH reduced nicotinamide adenine dinucleotide, RFU relative fluorescence units, RT room temperature, TN thermoneutrality, UCP1 uncoupling protein 1, IMM inner mitochondrial membrane. Statistical test: Two-tailed Student's t-test (B, E) and one-way ANOVA followed by LSD post hoc test (G, H, J, K). Data are expressed with individual values and mean ± SD superimposed. The exact *p*-value is displayed when $p < 0.05$. Source data are available online for this figure.

once norepinephrine (NE) was added to the media, suggesting the depolarizing effects of UCP1 outcompete the increase in MMP induced by oligomycin. Of note, NE had no effects on MMP when FCCP was present. These data suggest that our experimental approach allows detecting variations in MMP in intact cells.

Then, we tested the effects of IF1 silencing on MMP. First, we confirmed the efficacy of *Atp5if1* siRNA transfection by observing ~80% reduction in *Atp5if1* mRNA levels (Fig. 2A) as well as lower IF1 protein levels (Fig. 2B) compared to cells transfected with non-

targeting control siRNA (si*Scrambled*). To mimic cold exposure in vitro, we stimulated differentiated adipocytes with NE for 30 min before measuring MMP. As a control of our NE treatment, we determined p38-MAPK phosphorylation levels and as expected, NE stimulation increased phosphorylation of p38-MAPK, regardless of IF1 levels (Fig. 2C,D), suggesting that IF1 silencing does not exert a significant influence over adrenergic signaling. While we observed a mild reduction in MMP upon adrenergic stimulation, IF1 knock-down did not interfere with this effect (Fig. 2E). These results

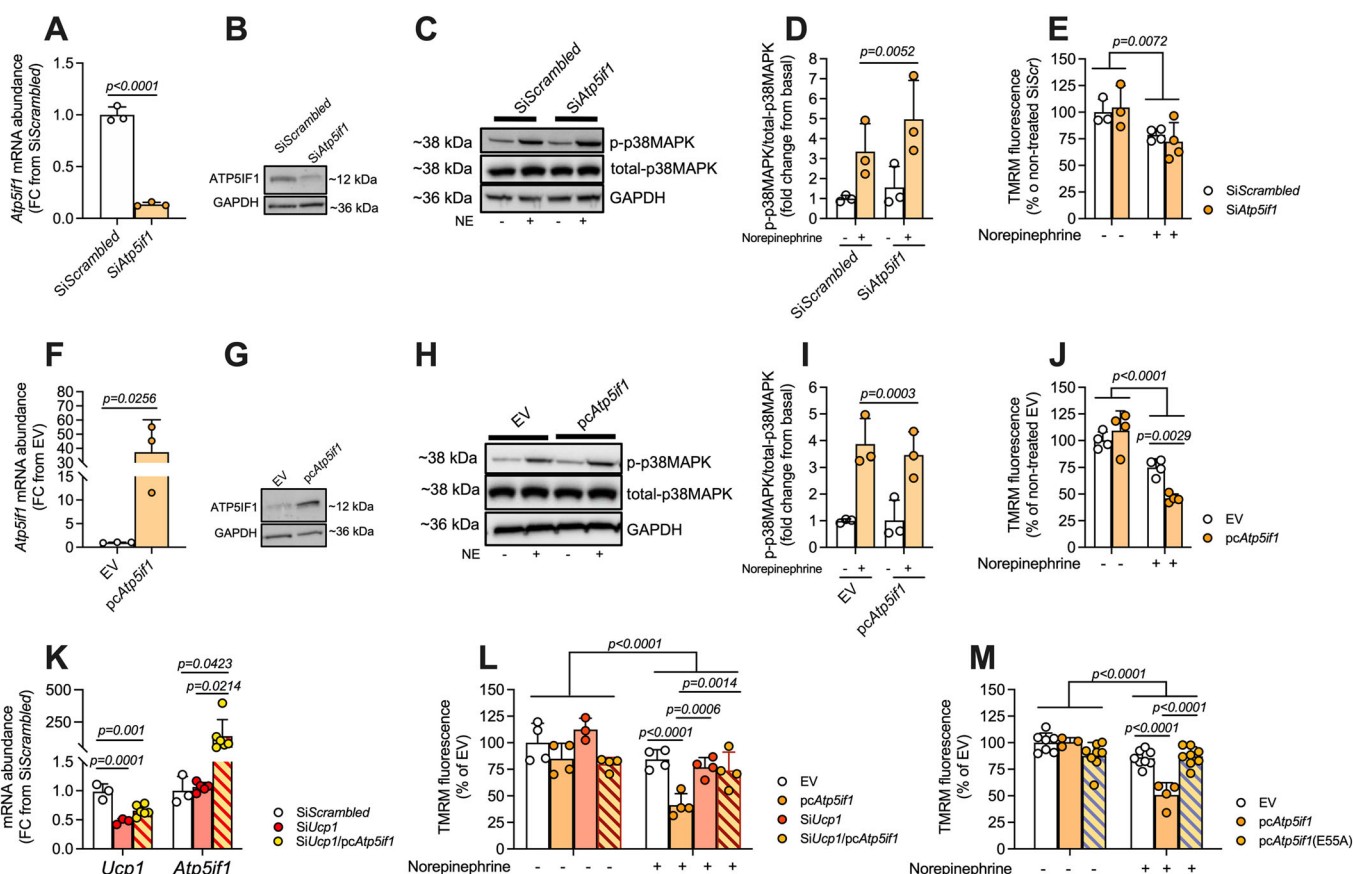

**Figure 2. IF1 modulates mitochondrial membrane potential upon adrenergic stimulation.**

(A) *Atp5if1* mRNA and (B) protein levels following Atp5if1 knockdown in differentiated WT1 brown adipocytes (three technical replicates). (C) Representative immunoblots and (D) quantification of p-p38-MAPK in non-stimulated and norepinephrine (NE)-stimulated (10 μM for 30 min) WT1 cells in which *Atp5if1* was knocked down (siAtp5if1) or in scramble controls (siScrambled) (three technical replicates from three biological experiments). (E) Relative TMRM fluorescence at baseline and upon NE stimulation in IF1-knocked down adipocytes (three/four technical replicates). (F) *Atp5if1* mRNA and (G) protein levels following IF1 overexpression in WT1 brown adipocytes (three technical replicates). (H) Representative immunoblots and (I) quantification of p-p38-MAPK in non-stimulated and NE-stimulated (10 μM for 30 min) WT1 cells in which IF1 was overexpressed (pcAtp5if1) or an empty vector control (EV) (three technical replicates from three biological experiments). (J) Relative TMRM fluorescence at the baseline and following NE stimulation (three technical replicates). (K) *Ucp1* and *Atp5if1* mRNA levels following *Ucp1* knockdown, IF overexpression, or both (three/six technical replicates). (L) Relative TMRM fluorescence at baseline and upon NE stimulation in *Ucp1* knocked down/IF1 overexpressing WT1 cells (four technical replicates). (M) Relative TMRM fluorescence at baseline and following NE stimulation in cells overexpressing mutant IF1(E55A) (four/six technical replicates). Atp5if1 ATP synthase inhibitory factor subunit 1, GAPDH glyceraldehyde-3-phosphate dehydrogenase, TMRM tetramethylrhodamine methyl ester, FC fold change, FCCP Carbonyl cyanide-p-trifluoromethoxyphenylhydrazone, p38MAPK p38 mitogen-activated protein kinase, NE norepinephrine, UCP1 Uncoupling protein 1, EV empty vector. Statistical test: Two-tailed Student's t-test (A, D, F, I), one-way ANOVA followed by LSD post hoc test (K, M), and two-way ANOVA followed by LSD post hoc test (E, J, L). Data are expressed with individual values and mean ± SD superimposed. Experiments were replicated in at least 3 independent experiments. The exact *p*-value is displayed when *p* < 0.05. Source data are available online for this figure.

demonstrate that loss of IF1 does not affect MMP in NE-stimulated brown adipocytes. Of note, despite a great increase in *Ucp1* mRNA levels, we did not observe modulation of IF1 by NE treatment in vitro in either mRNA or protein level (Fig. EV2B,C), confirming that the effect of NE over MMP under the in vitro condition is not linked to downregulation of IF1. We also noticed that the ATP synthase/IF1 ratio is greater in primary differentiated adipocytes compared to BAT (Fig. EV2D), indicating the impact of IF1 modulation is likely not similar between these two systems, and a direct comparison of these findings warrants caution.

Hence, to determine the influence of higher levels of IF1 on MMP upon NE stimulation of differentiated brown adipocytes, we performed gain-of-function experiments. Transfection with a vector carrying *Atp5if1* cDNA led to higher IF1 mRNA (Fig. 2F)

and protein (Fig. 2G) levels in differentiated mouse brown adipocytes. We confirmed higher p38-MAPK phosphorylation levels in NE-stimulated cells compared to non-stimulated controls (Fig. 2H) and found no apparent impairment in adrenergic signaling caused by IF1 overexpression (Fig. 2I). However, upon adrenergic stimulation, cells overexpressing IF1 showed a more pronounced drop in MMP (Fig. 2J). These results indicate that IF1 overexpression sensitizes the cell to decrease MMP further compared to control cells when stimulated with NE regardless of the absence of changes in adrenergic signaling (i.e., p-p38MAPK) levels). This is consistent with a possible inhibition of the capacity of ATP synthase to operate in the reverse mode, resulting in a lower contribution of ATP synthase to MMP and, consequently, greater depolarization of mitochondria. As UCP1 activation in brown

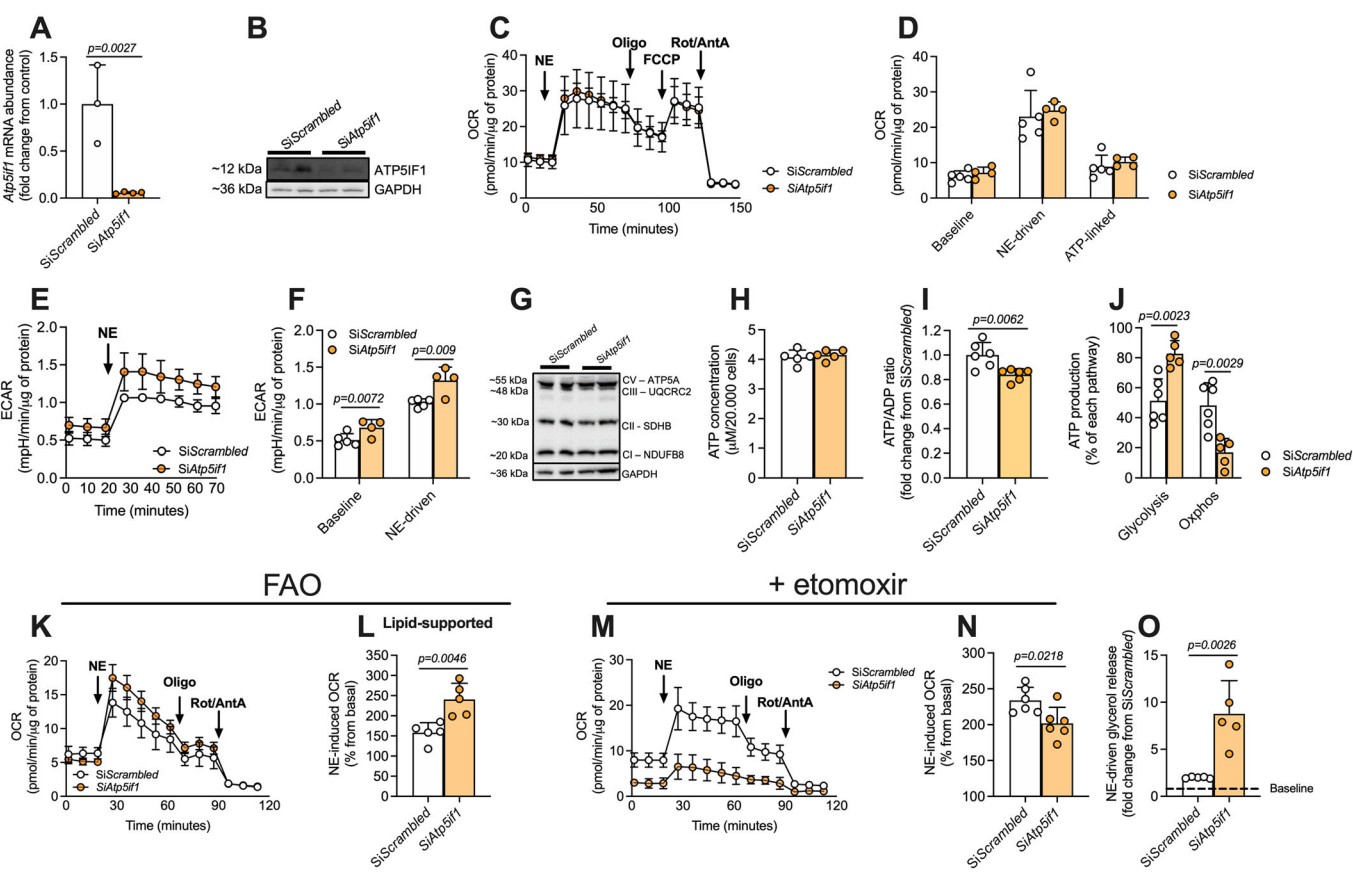

**Figure 3. IF1 knockdown induces mitochondrial lipid oxidation in brown adipocytes.**

(A) *Atp5if1* mRNA (three technical replicates) and (B) protein levels (two biological experiments). (C) Representative plot and (D) quantification of oxygen consumption rate following NE stimulation in primary brown adipocytes upon IF1 knockdown (si*Atp5if1*) or scramble control (si*Scrambled*) (four/five technical replicates). (E) Representative plot and (F) quantification of extracellular acidification rate in these cells (four/five technical replicates). (G) Representative immunoblots of OxPhos subunits in primary brown adipocytes (two biological experiments). (H) Cellular ATP content, (I) ATP/ADP ratio and (J) relative ATP production from glucose (five/six technical replicates). (K) Representative plot of fatty-acid oxidation-supported respiration (100 μM palmitate) and (L) quantification of NE-driven oxygen consumption (five technical replicates). (M) Representative plot and (N) NE-induced respiration in the presence of etomoxir (five technical replicates). (O) NE-driven glycerol release (five technical replicates). Atp5if1 ATP synthase inhibitory factor subunit 1, GAPDH glyceraldehyde-3-phosphate dehydrogenase, OCR oxygen consumption rate, NE norepinephrine, Oligo oligomycin, FCCP carbonyl cyanide-p-trifluoromethoxyphenylhydrazone, Rot rotenone, AA antimycin A, ATP adenosine triphosphate, ECAR extracellular acidification rate, OxPhos oxidative phosphorylation, CI complex I, CII complex II, CIII complex III, CV complex V, FAO fatty-acid oxidation. Two-tailed Student's t-test (A, D, F, H, I, J, L, N, O). The exact *p*-value is displayed when $p < 0.05$. Data are expressed with individual values and mean ± SD superimposed. Findings were replicated in at least 3 independent experiments. Source data are available online for this figure.

adipocytes is expected to influence MMP, we investigated the necessity of UCP1 for this mechanism to take place. For that, we silenced *Ucp1* mRNA in differentiated brown adipocytes and overexpressed IF1 at the same time (Fig. 2K). While *Ucp1* knockdown did not cause any major effect on MMP at baseline conditions or upon adrenergic stimulation, the reduction in MMP observed in cells overexpressing IF1 was abrogated when *Ucp1* was silenced (Fig. 2L). In summary, our data supports the notion that activation of UCP1 following adrenergic stimulation is a prerequisite for the modulatory effects of IF1 on MMP.

We next sought to determine whether the effect of IF1 overexpression was specifically related to its binding to ATP synthase. To test this, we overexpressed a mutant IF1 protein harboring an E55A substitution that renders the protein unable to interact with the ATP synthase (Fig. EV2E). Upon adrenergic stimulation, overexpression of IF1, but not E55A IF1, reduced

MMP in differentiated brown adipocytes (Fig. 2M). Altogether, our data show that high levels of IF1 in brown adipocytes result in mitochondria that cannot sustain MMP upon adrenergic stimulation, and this mechanism is dependent on UCP1.

## IF1 silencing potentiates mitochondrial lipid oxidation in brown adipocytes

Next, we investigated the metabolic implications of IF1 silencing in mouse primary brown differentiated adipocytes. We confirmed IF1 silencing by observing lower IF1 mRNA (Fig. 3A) and protein (Fig. 3B) levels in the cells upon transfection with *Atp5if1* siRNA. We then determined the oxygen consumption of these cells following NE stimulation. At first, we did not observe any differences in mitochondrial respiration (Fig. 3C,D). It has been suggested that lipolysis caused by adrenergic stimulation could

drive mitochondrial uncoupling in a UCP1-independent manner in non-buffered media (Li et al, 2014), which could represent a caveat in our experiments. Therefore, we repeated the experiment with the addition of 2% fatty acid-free BSA to buffer the excess of lipids caused by NE-induced lipolysis. However, we still could not detect any effect of IF1 knockdown on mitochondrial oxygen consumption at baseline or in response to adrenergic stimulation (Fig. EV3A,B). Of note, in the presence of 2% fatty acid-free BSA, we noticed IF1 knockdown in thermogenic adipocytes increased maximal activity of the electron transport chain (Fig. EV3A,B), suggesting an interaction between the absence of IF1 and lipids controlling respiration independently of ATP synthase and/or UCP1. Nevertheless, we detected higher acidification rates in IF1-deficient cells at baseline and after NE stimulation compared to control cells transfected with scramble siRNA (Fig. 3E,F), indirectly suggesting a greater reliance on aerobic glycolysis under these conditions. These differences were not explained by changes in oxidative phosphorylation (OxPhos) subunit levels (Fig. 3G) or by differences in total cellular ATP levels (Fig. 3H). However, ATP/ADP ratio was lower in IF1 knockdown adipocytes compared to scramble controls (Fig. 3I), suggesting cells with reduced IF1 levels experience mild energetic stress. The reduction in ATP/ADP ratio is one of the most important inducers of glycolysis (Kemp and Gunasekera, 2002; Schormann et al, 2019), supporting the notion that an increase in glycolysis could be a compensatory mechanism to sustain ATP levels in IF1 silenced cells. To test this hypothesis, we dissected the contribution of substrates to ATP synthesis using a reductionist, one-substrate approach. For that, we deprived cells of any other exogenous substrate and measured respiration and acidification rates after the injection of glucose to estimate cellular ATP production. We observed that cells, in which IF1 was silenced, generated more than 75% of ATP from aerobic glycolysis and less than 25% from mitochondrial respiration (Fig. 3J), whereas the source of ATP in control cells was roughly ~50% coming from aerobic glycolysis and ~50% from mitochondrial respiration. It is important to highlight that the estimation of ATP production in live cells by measuring $O_2$ consumption and extracellular acidification rates does not consider proton generation from other biological processes (e.g., tricarboxylic acid cycle or NAD(P) processes) or UCP1-dependent uncoupling that, despite the absence of adrenergic stimuli, are likely to be present. Therefore, interpretations of this experiment should be done carefully.

These results prompted us to investigate whether brown adipocytes with reduced IF1 levels display altered substrate preference to sustain normal overall respiration. As the measurement of whole-cell oxygen consumption does not differentiate between the substrates being used, we hypothesized that the reduced mitochondrial glucose utilization for oxidative phosphorylation in IF1-deficient cells could be paralleled by greater mitochondrial lipid oxidation, a process that uses more oxygen to generate ATP in comparison to carbohydrate oxidation. To test this hypothesis, we performed mitochondrial respiration analysis in brown adipocytes in the presence of 100 μM palmitate while in the absence of any other exogenous substrates. Using this experimental design, we observed that adipocytes, in which IF1 was silenced displayed higher respiration upon NE stimulation compared to control cells (Fig. 3K,L).

To evaluate the extent to which IF1-deficient brown adipocytes depend on lipids to sustain respiration, we measured cellular respiration using cell culture media containing glucose, pyruvate,

and glutamine in the presence of etomoxir, an inhibitor of carnitine-palmitoyl transferase 1, the rate-limiting step for lipid utilization by mitochondria. In the presence of etomoxir, mitochondrial respiration as well as NE-driven respiration was roughly 25% lower in IF1-deficient cells compared to controls (Fig. 3M,N). Of note, the metabolic remodeling observed after IF1 silencing resulted in higher lipolytic capacity (Fig. 3O), as observed by greater glycerol release upon adrenergic stimulation. Despite the increased potential to mobilize lipids through lipolysis, lipid content was similar between non-stimulated IF1-knockdown and Scrambled-control adipocytes (Fig. EV3C). Altogether, these data demonstrate that reducing IF1 levels primes mitochondria to utilize more lipids, thus supporting NE-induced uncoupling and mitochondrial oxygen consumption rate, while a compensatory mechanism seems to increase aerobic glycolysis to sustain cellular ATP levels.

## IF1 overexpression induces a quiescent-like state in brown adipocytes

As we demonstrated that higher levels of IF1 promote MMP reduction upon NE stimulation, we transiently overexpressed IF1 in differentiated primary brown adipocytes to test the impact of such manipulation on mitochondrial bioenergetics in mature cells. We first confirmed the overexpression of IF1 upon the transfection protocol by measuring *Atp5if1* mRNA (Fig. 4A) and protein (Fig. 4B) levels. We then measured mitochondrial respiration following acute NE stimulation. We observed a profound reduction in mitochondrial respiration as well as complete abrogation of NE-induced uncoupling in cells overexpressing IF1 compared to control cells transfected with the empty vector (Fig. 4C,D). Of note, the addition of 2% fatty acid-free BSA in the media did not abrogate the inhibitory effects of IF1 overexpression over brown adipocytes' mitochondrial respiration (Fig. EV4A,B). Intriguingly, we did not find any compensatory response in aerobic glycolysis in IF1-overexpressing cells (Fig. 4E,F). Moreover, we observed an almost 50% reduction in fatty acid-supported mitochondrial respiration at baseline and following NE stimulation in these cells (Fig. 4G,H). Overexpression of IF1 impaired mitochondrial respiration independently of respiratory complexes content (Fig. 4I). These data show that IF1 overexpression blunts mitochondrial respiration in primary brown adipocytes.

The reduction in mitochondrial respiration without any compensatory change in glycolytic activity led us to hypothesize that with high levels of IF1, the cells were transitioning into a quiescence-like state, resulting in overall lower metabolic activity. Thus, we estimated glucose-dependent ATP production from glycolysis and oxidative phosphorylation. We observed that total ATP production was almost 50% lower in cells overexpressing IF1 (Fig. 4J), and this was explained by lower oxidative phosphorylation-linked ATP production. Consequentially, cellular ATP content was ~50% lower in IF1 overexpressing cells compared to control cells (Fig. 4K). However, ATP/ADP ratio was not different between the groups (Fig. 4L), suggesting IF1-overexpressing cells did not experience energetic stress despite the reduction in mitochondrial OxPhos. Of note, we did not detect any differences in mRNA or protein yield, or in the lipolytic capacity of the cells overexpressing IF1 (Fig. EV4C–E), suggesting that these cells remained otherwise healthy. Hence, high levels of

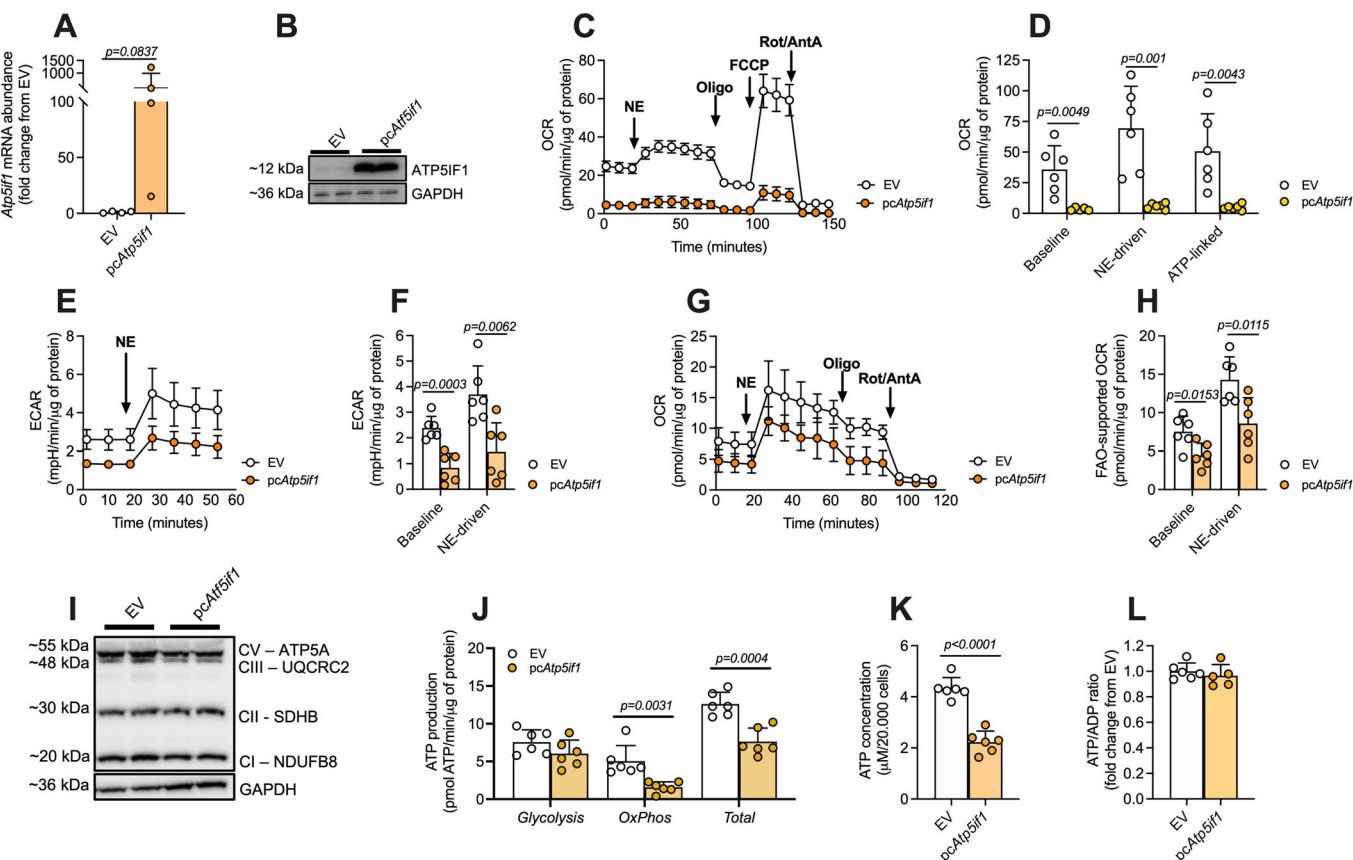

**Figure 4. IF1 overexpression blunts OxPhos and ATP production in primary brown adipocytes.**

(A) *Atp5if1* mRNA (four technical replicates) and (B) protein levels in primary brown differentiated adipocytes following Atp5if1 overexpression (two biological experiments). (C) Representative plot and (D) quantification of oxygen consumption rate following NE stimulation in primary brown adipocytes overexpressing IF1 (pcAtp5if1) or an empty vector (EV) (five/six technical replicates). (E) Representative plot and (F) quantification of extracellular acidification rate at baseline and upon NE stimulus (six technical replicates). (G) Representative plot of fatty-acid oxidation-supported respiration (100 µM palmitate) and (H) quantification of baseline and NE-driven oxygen consumption (six technical replicates). (I) Representative immunoblots of OxPhos subunits in primary brown adipocytes (two biological experiments). (J) Glycolytic, OxPhos, and total ATP production from glucose (six technical replicates); (K) total cell ATP content, and (L) ATP/ADP ratio (six technical replicates). Atp5if1 ATP synthase inhibitory factor subunit 1, GAPDH glyceraldehyde-3-phosphate dehydrogenase, EV empty vector, OCR oxygen consumption rate, NE norepinephrine, Oligo oligomycin, FCCP carbonyl cyanide-p-trifluoromethoxyphenylhydrazone, Rot rotenone, AA antimycin A, ATP adenosine triphosphate, ECAR extracellular acidification rate, Oxphos oxidative phosphorylation, CI complex I, CII complex II, CIII complex III, CV complex V, FAO fatty-acid oxidation. Two-tailed Student's t-test (A, D, F, H, J, K, L). The exact *p*-value is displayed when $p < 0.05$. Data are expressed with individual values and mean ± SD superimposed. Findings were replicated in at least 3 independent experiments. Source data are available online for this figure.

IF1 blunt mitochondrial respiration, ATP production, and NE-induced uncoupling in brown adipocytes.

## In vivo IF1 overexpression reduces mitochondrial respiration and limits adrenergic-induced NST in BAT

Having explored the effects of IF1 gain and loss-of-function in vitro, we interrogated the physiological effects of IF1 manipulation in vivo. For that, we employed two in vivo models of gain or loss-of-function. Given the temperature dependency on IF1 levels in BAT, we tested the effects of constitutive global IF1 knockout (IF1-KO) in thermoneutrality and BAT-restricted AAV-mediated IF1 overexpression below thermoneutrality. Compared to control mice, whole-body IF1-KO male mice did not show changes in body weight, food intake, baseline, or CL316,243-induced oxygen consumption adapted to room temperature or thermoneutrality (Fig. EV5A–H). These results mirrored our results obtained in

cultured adipocytes, in which IF1 knockdown in brown adipocytes did not change mitochondrial uncoupling or overall oxygen consumption in response to adrenergic stimulation when all substrates were present. Next, to test the complementary scenario of overexpressing IF1 when it is naturally downregulated by cold, we injected adeno-associated virus-carrying IF1 (AAV-Atp5if1) or a GFP-carrying vector (AAV-GFP) into BAT of adult mice housed at room temperature and exposed them to cold (4 °C) for 5 days (Fig. 5A). We confirmed our transduction protocol by observing roughly ~40-fold higher mRNA levels and ~2-fold higher protein levels of IF1 in the interscapular BAT depot (iBAT), respectively (Fig. 5B–D) while no changes in body weight or food intake were observed (Fig. EV6A,B).

Given that overexpression of IF1 in vitro led to a marked reduction in cellular respiration, we next tested the effects of IF1 overexpression on mitochondrial oxygen consumption in saponin-permeabilized BAT. We found that regardless of the substrate

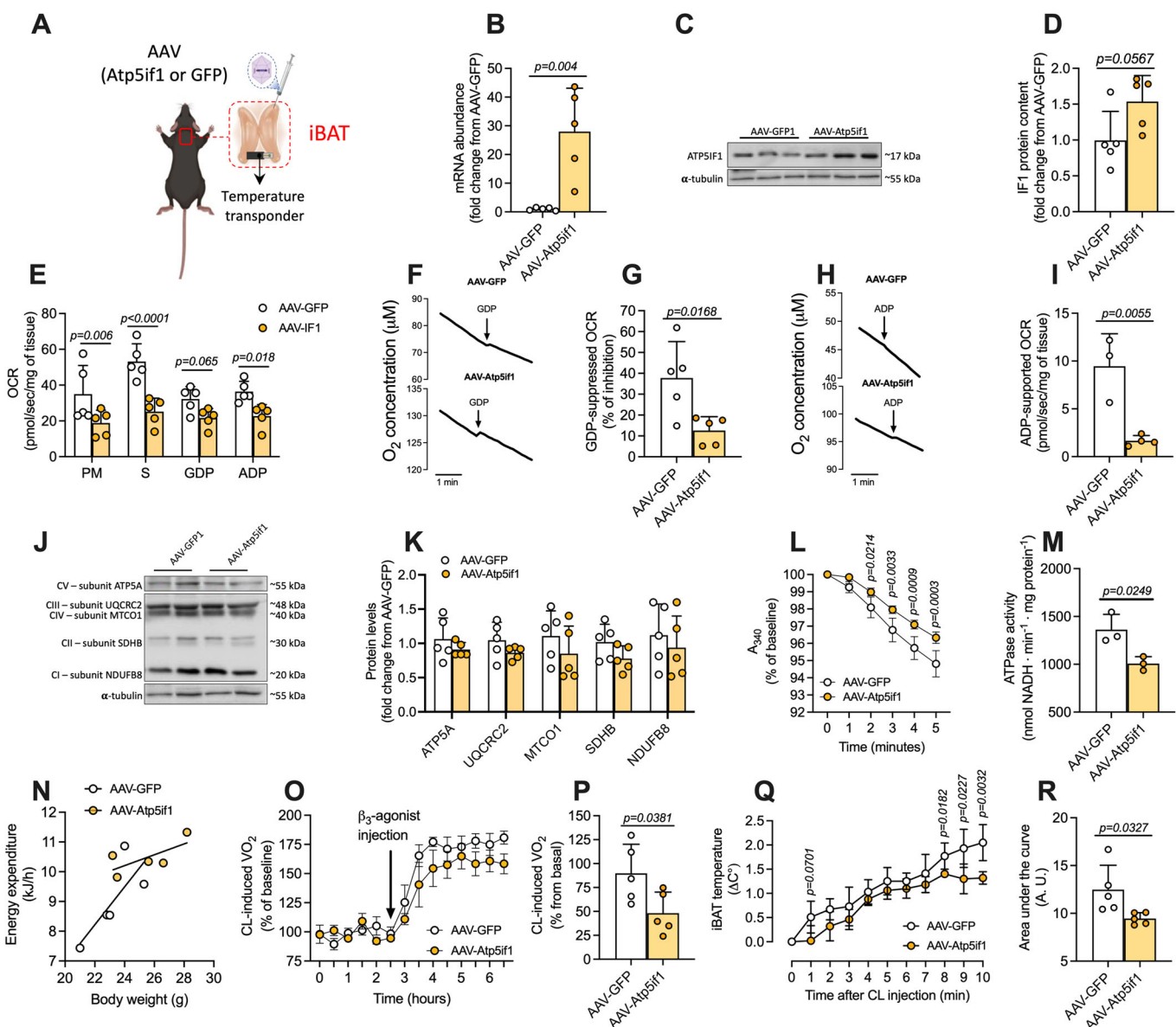

**Figure 5. In vivo IF1 overexpression in iBAT suppresses adrenergic-induced thermogenic response and mitochondrial respiration.**

(A) Model for AAV-induced IF1 overexpression in iBAT of male mice (created with BioRender). (B) IF1 mRNA levels in iBAT (five biological replicates); (C) representative immunoblot and (D) quantification of IF1 in iBAT (five biological replicates). (E) Mitochondrial oxygen consumption in the presence of pyruvate/malate (complex I), succinate (complex II), GDP (UCP1 inhibitor), and ADP (OxPhos stimulator) (five biological replicates). (F) Representative $O_2$ consumption changes induced by GDP and (G) quantification of % of inhibition caused by GDP (five biological replicates). (H) Representative mitochondrial respiration stimulated by ADP and (I) quantification of ADP-supported respiration in the presence of substrates and GDP (three/four biological replicates). (J) Representative and (K) quantification of immunoblot for mitochondrial complexes subunits (five biological replicates). (L) Representative trace and (M) quantification of ATP hydrolytic activity of ATP synthase of iBAT overexpressing IF1 (three biological replicates). (N) Correlation between body weight and energy expenditure (kJ/h) at 22 °C (five biological replicates). (O) Real-time recording of in vivo CL-316,24-induced oxygen consumption and (P) average of 3 h following injection (five biological replicates). (Q) Variation of iBAT temperature over 10 min and (R) area under the curve of CL-316,243-induced iBAT temperature (five biological replicates). iBAT interscapular brown adipose tissue, IF1 Inhibitor Factor 1, CL CL-316,243. P pyruvate, M malate, S succinate, GDP guanosine diphosphate, ADP adenosine diphosphate, OCR oxygen consumption rate, $A_{340}$ absorbance at 430 nm. Statistical test: Two-tailed Student's t-test. The exact p-value is displayed when p < 0.05. Data are expressed with individual values and mean ± SD superimposed. Findings were replicated in at least 3 independent experiments. Source data are available online for this figure.

offered, oxygen consumption was lower in BAT from AAV-Atp5if1 mice compared to AAV-GFP controls (Fig. 5E). Next, we added GDP, an inhibitor of UCP1, to estimate the contribution of uncoupled respiration in our assay. Interestingly, we found a ca. 40% reduction in mitochondrial respiration in BAT from AAV-

GFP mice while the effect of GDP in mitochondria from IF1 overexpressing mice was ca. 70% lower (Fig. 5F,G). Together, these data indicate hampered uncoupled respiration in BAT of AAV-Atp5if1 mice. Once UCP1 activity is inhibited, we added ADP into the chamber to stimulate respiration in a complex V-dependent

manner (i.e., OxPhos). By doing that, while in BAT from AAV-GFP mice ADP increased respiration by ~10 pmol/seg/mg of tissue, the capacity of ADP to drive respiration was completely suppressed in BAT derived from AAV-Atp5if1 mice (Fig. 5H,I). Of note, we did not find differences in OxPhos subunit contents (Fig. 5J,K) following IF1 overexpression in iBAT. In agreement with ADP-driven respiration, ATP hydrolytic activity in iBAT homogenates was lower in AAV-Atp5if1-injected compared to AAV-GFP-injected mice (Fig. 5L,M). Altogether, overexpression of IF1 in iBAT of adult mice results in lower respiratory capacity, lower uncoupled respiration, and lower ATP synthase hydrolytic activity.

While mice overexpressing IF1 within iBAT showed a slight increase in whole-body oxygen consumption during the light phase that was not explained by greater voluntary physical activity (Fig. EV6C,D), correlation analysis did not show any significant differences in overall energy expenditure when plotted against body weight between AAV-GFP and AAV-Atp5if1 mice (Fig. 5N). Although iBAT temperature was slightly higher in AAV-Atp5if1 mice compared to AAV-GFP at RT (Fig. EV6E), both groups showed similar iBAT temperature, body weight, and food intake variation during the 5 days of cold exposure (Fig. EV6E–G), suggesting IF1 overexpression in iBAT does not interfere with cold adaptation of mice (Fig. EV6E,F). Given the involvement of several complementary mechanisms to support NST, to specifically investigate the role of IF1 in iBAT, we assessed whole-body oxygen consumption and iBAT temperature in mice injected with CL316,243 (Collins, 2022). Interestingly, we found in vivo mitochondrial uncoupling induced by acute CL-316,243 injection was diminished in animals overexpressing IF1 in iBAT (Fig. 5O,P). Similarly, we noticed the increase of iBAT temperature after CL316,243 injection was lower in AAV-Atp5if1 injected mice compared to AAV-GFP controls (Fig. 5Q,R). Altogether, our data indicate overexpression of IF1 suppresses mitochondrial respiration, ATP synthase function, and suppresses the thermogenic response of iBAT.

## Discussion

Mitochondria in thermogenic adipocytes experience large fluctuations in metabolic rate depending on the ambient temperature. Sustaining MMP, especially considering the high activity of UCP1, is an important aspect of NST whose mechanism has not been fully elucidated. Here we show that IF1, an inhibitor of mitochondrial ATP synthase, is downregulated in cold-adapted BAT, allowing greater ATP synthase hydrolytic activity by operating in the reverse mode. In vitro, we found that IF1 overexpression in brown adipocytes makes mitochondria unable to sustain the MMP upon adrenergic stimulation and blunts mitochondrial respiration while IF1 knockdown phenocopies features of the metabolic adaptation of BAT to cold. Finally, in vivo IF1 overexpression in iBAT of mice lowers mitochondrial respiration and suppresses adrenergic response. Hence, a reduction of IF1 levels in BAT during cold exposure is a mechanism allowing proper bioenergetic adaptation during NST.

Yamada et al (1992) showed that in BAT from cold-adapted rats, the ATP synthesis capacity of mitochondria is preserved, whereas ATP hydrolysis is increased by almost 6-fold (Yamada et al, 1992). At the time, the authors proposed the existence of an unknown mechanism capable of controlling the counterclockwise rotation (hydrolysis) of ATP synthase without affecting its clockwise activity (synthesis). Our results shed light onto the mechanism by which BAT mitochondria adapt to the metabolic challenge imposed by cold exposure. Agreeing with the prediction by Yamada et al and providing a mechanistic underpinning for the phenomenon, our data reveal that IF1 is downregulated in BAT following cold exposure, thus facilitating the hydrolytic activity of ATP synthase and, possibly, helping sustain MMP. Although UCP1 activation lowers MMP, its impact on mitochondrial matrix pH is less clear, therefore, our findings offer an additional mechanism for controlling ATP synthase hydrolytic activity (i.e., IF1 levels) other than the well-documented pH-dependent regulation of IF1 (Cabezon et al, 2000). The suppressive effects of IF1 on MMP are blunted once a mutated IF1(E55A), incapable of binding to ATP synthase, is used, further demonstrating the effects of IF1 through the binding of IF1 to ATP synthase. Of note, the lack of apparent phenotype in brown adipocytes when $Ucp1$ was silenced could be explained by residual $Ucp1$ gene expression or by the stimulation of ATP-consuming processes (e.g., $Ca^{2+}$ cycling, membrane transport) that can lower mitochondrial membrane potential in a UCP1-independent manner. Nevertheless, the mechanism by which IF1 modulates MMP in brown adipocytes seems to be dependent on UCP1 activation. Therefore, it is plausible to speculate that the reduction of MMP upon adrenergic stimulation in IF1 overexpressing brown adipocytes could be a result of an inability of the ATP synthase to pump protons back into the intermembrane space due to inhibition of its hydrolytic activity. An alternative hypothesis is that IF1 overexpression would elicit defects in electron transport chain activity that could not match the activity of UCP1 upon adrenergic stimulation, which would lead to the collapse of MMP when UCP1 is activated. This speculation is supported by lower mitochondrial respiration in the presence of FCCP or substrates feeding complex I and complex II once IF1 is overexpressed in vitro and in vivo, respectively. Of note, IF1 has been also reported to modulate mitochondrial cristae shape in intestinal cells (Domínguez-Zorita et al, 2023), potentially affecting electron transport chain activity. Nevertheless, the role of IF1 in mitochondrial bioenergetics and metabolism appears to be more complex than previously anticipated.

Activation of BAT is associated with an increase in energy expenditure in the tissue as well as on the whole-body level. At the tissue level, BAT activation increases both lipid and glucose uptake (Bartelt et al, 2011; Hankir and Klingenspor, 2018; Sponton et al, 2022). Although the full comprehension of substrates that support short and long-term NST in BAT is not fully understood (Park et al, 2023), $\beta_3$-adrenergic agonist activation leads to a marked reduction in respiratory exchange ratio (Politis-Barber et al, 2022), suggesting an overall increase in lipid oxidation. Interestingly, IF1 downregulation is sufficient to prime brown adipocytes in vitro toward lipid utilization. Given that β-oxidation produces a higher $FADH_2/NADH$ ratio compared to glucose oxidation, fatty acid oxidation has a more pronounced effect on the increase in oxygen consumption (FAD-supported P/O ratio = 1.5; NAD-supported P/O ratio = 2.5) than glucose. In support of greater mitochondrial lipid utilization, we observed the estimated contribution of glycolysis to ATP production was increased in IF1 knockdown cells, facilitated by a lower ATP/ADP ratio. These data suggest that brown adipocytes in which IF1 is downregulated rely more on

glycolysis as an ATP source while lipids feed tricarboxylic acid cycle to support uncoupled mitochondrial respiration.

The response of BAT to adrenergic stimuli is lower when IF1 is overexpressed in BAT of mice, suggesting downregulation of IF1 is necessary to promote BAT respiratory capacity and support the metabolic demand imposed by adrenergic signaling. Indeed, IF1 overexpression in vitro and in vivo markedly suppresses mitochondrial respiration. Considering the unchanged levels of OxPhos complexes in both models, we exclude the possibility that low respiration could be a result of diminished mitochondrial content. In vitro, IF1 overexpression does not appear to induce an overall mitochondrial dysfunction, but rather pushes the cells into a quiescent-like state. This hypothesis is supported by the absence of metabolic compensatory mechanisms commonly found in models of mitochondrial dysfunction, such as increases in glycolysis, glycolysis-supported ATP production, or ATP/ADP ratio (Zhou et al, 2022; Formentini et al, 2012). It is noteworthy that upon cold adaptation, BAT undergoes remodeling by cell proliferation, inhibition of apoptosis, higher protein and lipid synthesis, and overall tissue expansion (Nedergaard et al, 2019). It has been shown that IF1 manipulation profoundly affects cellular metabolism and adaptation to stress in other contexts (e.g., cancer cells, skeletal muscle, cardiomyocytes) (Zhou et al, 2022; Formentini et al, 2012; Sa and Formentini, 2012), suggesting that higher levels of IF1 in BAT help brown fat cells remain in a quiescent-like state when NST is unnecessary. Interestingly, we observed that the effects of IF1 overexpression in BAT were compensated and masked by other thermogenic mechanisms. For instance, inguinal WAT weight and *Ucp1* mRNA levels were greater in mice overexpressing IF1 in iBAT compared to AAV-GFP mice (Fig. EV6H,I) following 5 days of cold exposure, indicating greater recruitment of beige adipocytes in this remote fat depot when IF1 is overexpressed in iBAT.

It remains unclear, however, how IF1 is downregulated following cold exposure. Notably, 4 h of cold exposure or CL administration in mice decrease *Atp5if1* mRNA levels in BAT by almost 50%, suggesting that the downregulation of IF1 may be controlled acutely at the transcriptional level. It has also been shown that immediate early response 1 (IEX1) targets IF1 for degradation (Shen et al, 2009). While IEX1 KO mice are protected from high-fat diet-induced insulin resistance through browning of adipose tissue, as far as we know, the response of IEX1 to cold exposure in BAT is unknown (Shahid et al, 2016). Therefore, proteolytic control of IF1 following cold exposure cannot be ruled out yet. Of note, proteomic analysis of BAT shows an increase of IF1 content following 4 and 24 h of cold exposure in mice (Forner et al, 2009); therefore, future studies should seek to conciliate such differences to better understand how IF1 is regulated under different metabolic contexts. Furthermore, exploring the non-canonical roles of IF1, including its regulation by soluble PKA (García-Bermúdez et al, 2015), involvement in cell proliferation, apoptosis, and differentiation (Formentini et al, 2012; Esparza-Moltó et al, 2017), alongside its impact on mitochondrial morphology (Domínguez-Zorita et al, 2023), is warranted, as the downregulation of IF1 with prolonged cold exposure may indicate broader alterations linked to adipose tissue remodeling. In addition, our analyses were performed in the whole homogenate or intact cells, therefore, ignoring possible subcellular changes in IF1 localization. Given that two different populations of mitochondria have been reported in thermogenic adipocytes (Benador et al,

2018), it would be interesting in the future to address IF1 modulation in these distinct populations to further expand our knowledge on the role of IF1 in mitochondrial bioenergetics.

In summary, the decline in IF1 levels observed in BAT during cold exposure serves as an adaptive mechanism, facilitating the remodeling of mitochondrial and cellular metabolism to support NST. We propose that this mechanism enables brown adipocytes to maintain MMP when UCP1 and NST are chronically activated. As a result, modulation of ATP synthase activity by IF1 downregulation is an additional mechanism to support BAT adaptation to NST.

## Methods

### Animals and indirect calorimetry

All experiments were performed following institutional guidelines and approved by the Animal Ethical Committee at the University of Campinas (5929-1/2021) and the government of Upper Bavaria, Germany (Protocol number 02-21-160) and performed in compliance with German Animal Welfare Laws. We followed the ARRIVE guidelines (Percie Du Sert et al, 2020). For in vivo experiments, male C57BL/6J mice (12–15 weeks old) were randomly divided into room temperature (22 °C) or cold exposure (4 °C) groups for 3 or 5 days. After each experimental protocol, mice were anesthetized with ketamine and xylazine (360 mg/kg and 36 mg/kg, respectively) and once the absence of reflex was confirmed, tissues were harvested followed by cervical dislocation. All animals were housed on 12 h light:dark cycle with 24-h access to chow diet and water ad libitum (diet PRD00018 Nuvilab, Suzano, Brazil).

To generate mice lacking IF1, encoded by the Atpif1 gene, we acquired Atpif1 knockout mice (Atpif1tm1a(EUCOMM)Wtsi) from the European Mouse Mutant Archive, a component of the International Mouse Phenotyping Consortium. This Atpif1 tm1a allele is flanked by FRT and loxP sites. To produce whole-body IF1 knockout (IF1 KO) mice, offspring carrying the Atpif1 tm1a allele were crossed with mice expressing a global Cre-deleter strain (β-actin Cre) on a C57BL/6 background to delete exon 3 and generate the *Atpif1tm1b* allele (KO). Whole-body IF1 KO male mice were kept at either room temperature (23 °C) for 3–4 weeks or thermoneutrality (28 °C) for two weeks. Food intake and body weight were determined weekly and after the end of the adaptation period, whole-body oxygen consumption as well as CL-316,243-induced adrenergic (1 mg/kg of body weight) response was measured using indirect calorimetry. Resting and CL-316,243-induced oxygen consumption was indirectly measured using Sable Systems Promethion Indirect Calorimetry System (Kotschi et al, 2022).

For animals with IF1 overexpression in BAT (see protocol below), after 3 weeks of surgery, resting oxygen consumption ($VO_2$) and carbon dioxide production ($VCO_2$) were monitored in metabolic cages (Columbus Instruments, Columbus, OH, USA) at 22 °C (Brunetta et al, 2020). $β_3$-adrenergic agonist-mediated energy expenditure was measured for 3 h or iBAT-dependent heat production for 10 min after CL316,243 (1 mg/kg of body weight) intraperitoneal injection during the light cycle of the animal at 22 °C. All experiments were performed in non-blinded fashion.

## AAV production and IF1 overexpression in BAT

The AAV plasmid for IF1 overexpression was acquired from Origene Technologies GmbH (Reference CW309970). AAV packaging, titration, and injection into BAT were performed according to a previous study (Valdivieso-Rivera et al, 2023). Briefly, AAVs were produced by triple transfection of Adeno-X 293 cells (Takarabio) with the targeting vector plasmid, the pAdDeltaF6 plasmid (Addgene #112867), and the pAAV2/8 plasmid (Addgene # 112864) using polyethylenimine (1 µg/µl) (Sigma-Aldrich, 408727). The cells were collected by scratching the plates 72 h post-transfection and filtered using an Amicon Ultra-0.5 centrifugal filter (Merck Millipore, UFC510024). Extraviral DNA was removed by digestion with DNase I (Thermo Scientific, EN0521) and the viral particles were released through lysis of the cells. Finally, the virus titer was quantified using quantitative polymerase chain reaction (qPCR) with SYBR Green Master Mix (Thermo Scientific, 4309155) and primers targeting ITRs (Appendix Table S1).

Experiments with IF1 overexpression in BAT were carried out using male C57BL/6J mice, provided by Multidisciplinary Center for Biological Research (University of Campinas). Mice were 15–17 weeks old when adeno-associated virus (AAV) was injected into the interscapular BAT depot. Two weeks after AAV injection, indirect calorimetry was performed followed by acute intraperitoneal CL316,243 injection (described above) before exposing animals to cold exposure (4 °C) for 5 days. All animals were housed at the Institute of Biology animal facility on 12 h light:dark cycle with 24 h access to chow diet and water ad libitum (diet PRD00018 Nuvilab, Suzano, Brazil).

Mice were anesthetized with 1–2% isoflurane using a Vaporizer AI-100 (Insight Ltda). A 0.3–0.8 cm longitudinal incision was made in the skin at the interscapular region to expose the BAT. Thirty microliters of AAV ($2 \times 10^{11}$ vector genomes) were administered in both lobes (10 injections of 3 µL each in different anatomical spots) of the interscapular BAT depot of mice using a Hamilton syringe. Following the injections, IPTT-300 Temperature Transponders (PLEXX, 11059) were placed onto the interscapular BAT. After the surgery, mice were sutured with surgical strain and received ibuprofen diluted via drinking water (1 mg/ml) for 5 days. BAT temperature was monitored daily using the transponders.

## F$_o$F$_1$-ATP synthase hydrolytic activity

F$_o$F$_1$-ATPase activity was measured in total BAT homogenates using a spectrophotometric method adapted from a previous publication (Petrick et al, 2022). Briefly, 190 µL of reaction buffer containing 200 mM KCl, 20 mM HEPES, 10 mM NaN$_3$, 1 mM EGTA, 15 mM MgCl$_2$, and 10 mM phosphoenolpyruvate (pH 8) was added to a 96-well plate. Immediately before the reaction, 18 U/mL lactate dehydrogenase, 18 U/mL pyruvate kinase, 10 µL BAT homogenate, and 0.2 mM NADH were added to the well for a final volume of 0.2 mL. Assays were performed in triplicates at 37 °C and 340 nm wavelength. F$_o$F$_1$-ATPase synthase hydrolytic activity was measured after the addition of 5 mM ATP. The slope of NADH disappearance after 5 min of reaction was used to calculate F$_o$F$_1$-ATPase activity and averaged among the triplicates. To assure F$_o$F$_1$-ATPase synthase activity, we used wells with the addition of oligomycin 1 µg/mL (F$_o$-ATP synthase inhibitor), or in the absence of ATP or Mg$^{2+}$ (cofactor necessary for F$_o$F$_1$-ATPase function).

## Mitochondrial isolation

Interscapular BAT mitochondria were isolated using differential centrifugation as previously described (Brunetta et al, 2022). Tissues were harvested, minced in isolation buffer (100 mM sucrose, 100 mM KCl, 50 mM Tris-HCl, 1 mM KH$_2$PO$_4$, 0.1 mM EGTA, 0.2% BSA, and 1 mM ATP; pH 7.4), weighed, and manually homogenized using a Teflon pestle. Whole-tissue homogenate was centrifuged at $800 \times g$ for 10 min, resuspended in 4 mL of isolation buffer, and immediately spun at $5000 \times g$ for 5 min. The pellet was repeatedly resuspended in respiration buffer (0.5 mM EGTA, 3 mM MgCl$_2$·6H$_2$O, 60 mM K-lactobionate, 10 mM KH$_2$PO$_4$, 20 mM HEPES, 20 mM taurine, 110 mM sucrose, 1 g/L free acid-free BSA; pH 7.1) and pelleted at $10,000 \times g$ for 10 min. After protein quantification, 50 µg of mitochondrial protein was used to determine MMP.

## ATP-supported mitochondrial membrane potential (ΔΨ) estimation

MMP was determined using 5 µM safranin-O dye added to the reaction medium as previously described (Francisco et al, 2018). Briefly, isolated mitochondria from BAT of room temperature or cold-exposed mice were incubated in the presence of antimycin A and GDP to inhibit flux through respiratory complex III and UCP1, respectively. After signal stabilization, ATP was added to the media to stimulate the reverse mode of ATP synthase and membrane potential generation. Oligomycin was used as a control of specificity. Data are present as % of baseline signal obtained by safranin O fluorescence.

## Cell culture

Brown preadipocyte cells WT1, kindly donated by Dr. Brice Emanuelli (University of Copenhagen), were cultured at 37 °C, 5% CO$_2$, in DMEM GlutaMax growth medium (Thermo Fisher, cat. Num. 31966) containing 10% fetal bovine serum (FBS, Sigma-Aldrich) and 1% penicillin/streptomycin (Sigma-Aldrich) and differentiated as previously described (Willemsen et al, 2022). Briefly, upon confluence, cells were differentiated from mature adipocytes by the addition of 1 µM rosiglitazone (Cayman Chemicals), 1 nM triiodothyronine (T3) (Sigma-Aldrich), 850 nM human insulin (Sigma), 500 nM 3-isobutyl-1-methylxanthine (IBMX, Sigma-Aldrich), 1 mM dexamethasone (Sigma-Aldrich), and 125 nM indomethacin (Sigma-Aldrich) for 48 h, after which the medium was changed to a growth medium containing only rosiglitazone, T3, and Insulin with this medium being renewed every 2 days.

## Primary brown adipocytes culture

Primary brown adipocytes were obtained from interscapular brown adipose tissue of 6–8-week-old C57BL/6N male mice. After collection of intrascapular BAT, tissues were minced and digested using 2 mg/ml collagenase at 37 °C under continuous agitation. Then, cells were filtered using 100 µm and 70 µm cell strainers and cultured at 37 °C, 5% CO$_2$, in F12 media containing 10% FBS (Sigma-Aldrich), and 1% penicillin/streptomycin (Sigma-Aldrich) until confluence. Upon confluence, cells were induced to differentiation by the addition of 1 µM rosiglitazone (Cayman Chemicals), 1 nM T3 (Sigma-Aldrich), 850 nM human insulin

(Sigma), 500 nM IBMX (Sigma-Aldrich), and 1 mM dexamethasone (Sigma-Aldrich) for 48 h, after which the medium was changed to a growth medium containing only rosiglitazone, T3, and insulin with this medium being renewed every 2 days. Experiments in both cell lines were carried out on day 6 of the differentiation protocol. Acute norepinephrine treatment was carried out by diluting norepinephrine to a 10 μM final concentration and treating the cells for 30 min. After that, immunoblots, membrane potential, or cellular ATP content were determined.

### In vitro IF1 gain- and loss-of-function experiments

For the loss-of-function experiments, knockdown was achieved by using SMARTpool siRNA (Dharmacon). Transfection was performed on day 4 of differentiation using LipofectamineRNAiMAX transfection reagent (Thermo Fisher) and siRNA targeting *Atp5if1* and/or *Ucp1* at a concentration of 30 nM. SiScrambled was used as a control for silencing experiments. For the gain-of-function experiments, TOP10 competent bacteria were transformed by mixing 1 μl of Atp5if1_pcDNA3.1 +/C-(K)-DYK plasmids and keeping them for 30 s at 42 °C. After that, bacteria were grown at 37 °C for 1.5 h and then streaked onto an agar plate containing ampicillin (100 μg/ml) and grown overnight also at 37 °C. In the next day, a single colony was picked and inoculated into 5 mL LB medium with 100 μg/mL ampicillin. In the evening, 1 mL of the day culture was transferred into 400 mL LB medium with ampicillin for overnight culture at 37 °C, 200 rpm. Maxiprep was performed according to the manufacturer's instructions (NucleoBond® Xtra Maxi Plus EF, Macherey-Nagel), and, after elution, DNA concentration was determined with Nanodrop and diluted to 1 μg/μl. Cells were transfected with the plasmid using TransIT-X2 diluted in Opti-MEM I Reduced-Serum Medium according to the manufacturer's instructions. In the next day, cell medium was replaced and cells were incubated for another 24 h before the experiments. P-MXs-IF1(E55A) mutation was generated by David Sabatini's group and deposited at AddGene (cat. Number #85404). The details of the mutation generation can be found elsewhere (Chen et al, 2014).

### Mitochondrial membrane potential (ΔΨ) determination in intact cells

To determine mitochondrial membrane potential, 20,000 differentiated WT1 cells were seeded onto a well of a 96-well plate. Twenty-four hours after transfection, media was changed and kept for another 24 h. Then, cells were treated with 10 μM norepinephrine for 30 min and, after that, stained with 20 nM TMRM for 30 min (Abcam, ab228569) according to the instructions of the manufacturer. Following incubation, cells were washed with imaging buffer (Abcam, ab228569) and imaged using a Tecan plate reader at wavelength excitation/emission = 548/575 nm, respectively. Fluorescence was normalized relative to non-treated scrambled or empty vector cells in the absence of norepinephrine.

### ATP levels and ATP/ADP ratio determination

Resting cellular ATP levels were determined in differentiated WT1 cells by luminescence (Luminescent ATP detection kit, Abcam, ab113849). Cells were transfected as previously described.

Thereafter, media was washed off and cells were incubated with lysis buffer for 5 min under constant agitation (300 rpm). Then, ATP detection reaction buffer was added, and the samples were read in a Tecan plate reader according to the instructions of the manufacturer. Luminescence values were plotted against a standard curve provided by the manufacturer. To estimate ADP content, 100 μM dCTP (Sigma-Aldrich, cat. Num. 11934520001) and 5 U/ml nucleoside 5-diphosphate kinase (Sigma-Aldrich, cat. Num. N2635) were added, and luminescence was read again after 10 min (Ruas et al, 2018).

### Cellular oxygen consumption and extracellular acidification rate

Mitochondrial respiration was measured using Seahorse Cell Mito Stress Test (Agilent) with some adjustments to the manufacturer's protocol. Briefly, primary differentiating brown adipocytes were seeded onto a 24-well Seahorse plate on the fourth day of differentiation. After transfection, culture medium was replaced with Seahorse medium (XF DMEM pH 7.4, 10 mM glucose, 1 mM pyruvate, 2 mM L-glutamine). To determine fatty-acid-supported respiration, Seahorse medium was supplemented with 100 μM palmitate dissolved in 1% free-fat acid-BSA (Sigma-Aldrich, cat. Num. A8806) while other substrates (i.e., glucose, pyruvate, glutamate) were not added to the respiration media. Cells were incubated for 60 min at 37 °C without $CO_2$ before being placed in the Seahorse Analyzer Xfe24 instrument. When indicated, 10 μM etomoxir (Sigma-Aldrich, cat. Num. 236020) was added in this step into the medium. In the assay, the cells were treated with norepinephrine (Sigma-Aldrich, cat. Num. A0937) (final concentration in the well was 1 μM), oligomycin (1 μM), FCCP (2 μM), and rotenone/antimycin A (0.5 μM) (Sigma-Aldrich, cat. Num. 75351, C2920, 557368, and A8674, respectively). The reagents were mixed for 3 min, followed by 3 min incubation, and 3 min measurement. Total protein was measured for normalization using BCA assay (ThermoFisher, cat. Num. 23225) according to the manufacturer's instructions. To test the effects of FFA-induced by lipolysis on mitochondrial uncoupling independent of UCP1, mitochondrial respiration was also determined in Seahorse medium (as described before) with the addition of 2% fatty acid-free BSA, as previously published (Li et al, 2014).

Mitochondrial respiration in BAT from IF1 overexpressing mice was determined in saponin-permeabilized adipose tissue in an Oxygraph high-resolution respirometer chamber with 2 mL MiR05 at 37 °C as previously described with minor modifications (Brunetta et al, 2020). Briefly, BAT was excised and immediately placed in 1 ml of BIOPS (2.77 mM $CaK_2$-EGTA, 7.23 mM $K_2$-EGTA, 5.77 mM $Na_2$-ATP, 6.56 mM $MgCl_2 \cdot 6H_2O$, 15 mM $Na_2$-PCr, 20 mM imidazole, 0.5 mM dithiothreitol, and 50 mM NA). After that, a small piece was weighed (~2–4 mg of wet tissue) and minced with scissors. Tissues fragments were then transferred to 2 mL of MiR05 buffer (0.5 mM EGTA, 3 mM $MgCl_2 \cdot 6H_2O$, 60 mM K-lactobionate, 10 mM $KH_2PO_4$, 20 mM HEPES, 20 mM taurine, 110 mM sucrose, 1 g/L FA-free BSA; pH 7.1) and used to determine rates of oxygen consumption by high-resolution respirometry (Oroboros Oxygraph-2k, Innsbruck, Austria) in the presence of saponin (40 μg/mL). BAT mitochondrial $O_2$ consumption was tested by sequentially adding 5 mM pyruvate + 2 mM malate, 10 mM succinate; 2 mM GDP was used to inhibit uncoupling protein 1 (UCP1), and 2 mM ADP to test the contribution of oxidative

phosphorylation (OxPhos) after artificially coupling mitochondria with GDP.

## Estimation of glucose-dependent ATP production

Glycolytic and oxidative ATP supply rates were estimated from cellular oxygen consumption and medium acidification in an Agilent Seahorse XF Analyzer as described in full detail by others (Mookerjee et al, 2017), assuming that cellular energy metabolism was fueled exclusively by glucose and no proton leak is found either in the inner mitochondrial membrane during oxidative phosphirylation or in other mitochondrial processes (i.e., tricarboxylic acid cycle, NAD(P)(H) cycles). For that, we supplemented our media only with glucose. Sequential injections were made as follows: glucose (final concentration 10 mM), oligomycin (1 μM), and rotenone/antimycin A (0.5 μM). Glycolytic and oxidative phosphorylation ATP supply were calculated in the presence of glucose after subtracting oxygen consumption rate and extracellular acidification rate from oligomycin and rotenone/antimycin A injection. After the experiment, total protein was measured for normalization using BCA assay (ThermoFisher) according to the manufacturer's instructions.

## Glycerol release

Glycerol concentration in the media was used as a surrogate for lipolysis. Glycerol release was measured at baseline and upon adrenergic stimulation with norepinephrine as previously reported (Willemsen et al, 2022). Briefly, we used Free Glycerol Reagent (Sigma-Aldrich, F6428) and Glycerol standard solution (Sigma-Aldrich, G7793) to measure free glycerol concentrations in the cell culture supernatant. We replaced the culture medium and then collected the new one after 90 min (baseline condition); then we replaced it with a new culture medium in the presence of 1 μM norepinephrine for another 90 min. The kit was used according to the manufacturer's instructions. Fold change was calculated by the ratio between norepinephrine-stimulated glycerol release and the baseline glycerol values within each well.

## Oil-Red-O staining

We used Oil-Red-O (ORO) staining to measure lipid 1earts1i in adipocytes. Cells were washed with DPBS (Gibco), and fixed in zinc formalin solution (Merck) for 60 min at room temperature. ORO working solution was prepared (60% v/v Oil-Red-O solution (Sigma-Aldrich), 40% v/v $H_2O$) and filtered twice through a funnel with filter paper. After the incubation time, zinc 1earts1is was carefully removed and cells were washed with $H_2O$. 60% isopropanol was added and incubated for 5 min. Then, isproanol was aspirated and ORO working solution was added and incubated for 5 min. Cells were again rinsed with $H_2O$ and then counterstained with hematoxylin for 1 min. After rinsing with water, wells were kept wet until 1earts1 under the microscope to prevent lipid droplet disruption.

## RNA extraction, cDNA synthesis, and qPCR

Total RNA extraction was performed using the NucleoSpinRNA kit (Macherey-Nagel) as specified by the manufacturer. Cells were lyzed in TRIzol (Thermo Fisher) using and mixed with chloroform at 1:5 v/v ratio (chloroform:TRIzol), samples were then centrifuged, and the supernatant transferred into the purification columns of the NucleoSpinRNA kit. All further steps were executed as specified by the manufacturer. cDNA was synthesized with Maxima H Master Mix 5 (Thermo Fisher) using 500 ng of total RNA. Gene expression was evaluated by qPCR using PowerUpSYBR Green Master Mix (Thermo Fisher) according to the manufacturer's instructions. Primers are listed in Appendix Table S1. Expression was normalized to *Tbp* levels by the ΔΔct-method.

## Immunoblots

The samples were lysed in RIPA buffer [50 mM Tris (Merck), pH 8, 150 mM NaCl (Merck), 5 mM EDTA (Merck), 0.1% w/v SDS (Carl Roth), 1% w/v IGEPALCA-630 (Sigma-Aldrich), 0.5% w/v sodium deoxycholate (Sigma-Aldrich)] freshly supplemented with protease inhibitors (Sigma-Aldrich) in a 1:100 v:v ratio and phosStop phosphatase inhibitors (Roche). Cell lysates were centrifuged for 15 min (4 °C, 12,000 × g) and tissue lysates were centrifuged twice for 15 min, 6000 × g at 4 °C before the supernatant was collected. Protein concentrations were determined using the Pierce BCA assay (Thermo Fisher) according to the manufacturer's instructions. Protein samples were denatured with 5% v/v 2-mercaptoethanol (Sigma-Aldrich) for 5 min at 95 °C before they were loaded in gradient Bolt Bis-Tris gels (Thermo Fisher). After separation, proteins were transferred onto a 0.2 mm PVDF membrane (Bio-Rad) using the Trans-BlotTurbosystem (Bio-Rad) at 27 V, 1.4 A for 7 min. The membrane was blocked in Roti-Block (Roth) for 1 h at room temperature. The membranes were incubated overnight in primary antibody (Appendix Table S2) dilutions in 5% BSA-TBST at 4 °C. After washing with TBST [200 mM Tris (Merck), 1.36 mM NaCl (Merck), 0.1% v/v Tween20 (Sigma-Aldrich)], the membranes were incubated in secondary antibody (Santa Cruz) solutions (1:10,000 in Roti-block) for 1 h at RT. Membranes were washed in TBST and imaged using Super-Signal West Pico PLUS Chemiluminescent Substrate (Thermo Fisher) in a Chemidoc imager (Bio-Rad). Images were analyzed using ImageJ (www.imagej.nih.gov). Of note, we noticed a difference in the migration pattern of IF1 when using pre-made commercial gels (Thermo Fisher, cat. Num. NP0326BOX) compared to homemade ones (0.37 M Tris-Base, 15% acrylamide/Bis *v/v*, 1% SDS *v/v*, pH 8.8). When using pre-made commercial gels, IF1 appears between 12 and 15 kDa whereas in homemade gels, IF1 appears between 15 and 17 kDa. Given the same antibody against ATP5IF1 was used throughout the entire study, gel composition explains a slight change in the molecular weight appearance between our experiments without, nevertheless, affecting the capacity to detect IF1 in tissue or cell homogenates.

## Graphics

The graphical abstract and cartoons in Figs. 1 and 5 were created using BioRender.com.

## Statistical analysis

Data are shown as individually or mean ± standard deviation (SD). No statistical method was used to calculate sample size. Outliers

were removed when the observation was greater than 2 times the SD within the same group. All data was subjected to distribution tests before statistical comparisons were made. Comparisons between two groups were made using a two-tailed Student's t-test while comparisons of three groups were done by using one-way ANOVA followed by LSD post hoc test. When two levels were tested (i.e., treatment vs. IF1 manipulation) two-way ANOVA followed by LSD post hoc test was used. Post hoc tests were only applied once an interaction between conditions was found, otherwise, the *p*-value of the main effect of one of the two conditions is reported. Analysis was performed using GraphPad Prism (La Jolla, CA, USA). Statistically significant differences were indicated as the exact p-value when $p < 0.05$.

## Data availability

No large-scale data amenable to data repository deposition were generated in this study.

The source data of this paper are collected in the following database record: biostudies:S-SCDT-10_1038-S44318-024-00215-0.

## Peer review information

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

## Acknowledgements

We thank Dr. Elzira Saviani, Silvia Weidner, and Thomas Pitsch for their excellent technical assistance. HSB, SCCZ, FV, VF, RFC, and MAM were supported by the Fundação de Amparo à Pesquisa do Estado de São Paulo – FAPESP (2019/21852-1, 2022/00358-1, 2023/07882-0, 2021/08354-2, 2013/07607-8, 2017/17728-8, 2023/00229-0, 2022/13145-6, 2020/14725-0). MAM was funded by the Conselho Nacional de Desenvolvimento Científico e Tecnológico (CNPq) (310287/2018-9 and 306193/2022-1) and Coordenação de Aperfeiçoamento de Pessoal de Nível Superior – Brasil (CAPES) (88881.143924/2017-01). AB was supported by the Deutsche Forschungsgemeinschaft (SFB1123-B10 & SPP2306 BA4925/2-1), the Deutsches Zentrum für Herz-Kreislauf-Forschung DZHK, and the ERC Starting Grant PROTEOFIT.

## Author contributions

**Henver S Brunetta**: Conceptualization; Data curation; Formal analysis; Investigation; Visualization; Methodology; Writing—original draft; Project administration; Writing—review and editing. **Anna S Jung**: Data curation; Formal analysis; Investigation; Methodology; Writing—review and editing. **Fernando O Valdivieso-Rivera**: Data curation; Methodology; Writing—review and editing. **Stepheny C de Campos Zani**: Data curation; Methodology; Writing—review and editing. **Joel Guerra**: Data curation; Methodology; Writing—review and editing. **Vanessa O Furino**: Data curation; Methodology; Writing—review and editing. **Annelise Francisco**: Data curation; Methodology; Writing—review and editing. **Marcelo Berçot**: Data curation; Methodology; Writing—review and editing. **Pedro M Moraes-Vieira**: Data curation; Methodology; Writing—review and editing. **Susanne Keipert**: Conceptualization; Data curation; Writing—review and editing. **Martin Jastroch**: Conceptualization; Data curation; Writing—review and editing. **Laurent O Martinez**: Conceptualization; Data curation; Methodology; Writing—review and editing. **Carlos H Sponton**: Conceptualization; Data curation; Methodology; Writing—review and editing. **Roger F Castilho**: Conceptualization; Data curation; Methodology; Writing—review and editing. **Marcelo A Mori**: Conceptualization; Supervision; Funding acquisition; Writing—original draft; Writing—review and editing. **Alexander Bartelt**: Conceptualization; Supervision; Funding acquisition; Validation; Writing—original draft; Project administration; Writing—review and editing.

Source data underlying figure panels in this paper may have individual authorship assigned. Where available, figure panel/source data authorship is listed in the following database record: biostudies:S-SCDT-10_1038-S44318-024-00215-0.

## Funding

## Disclosure and competing interests statement

The authors declare no competing interests.

# Expanded View Figures

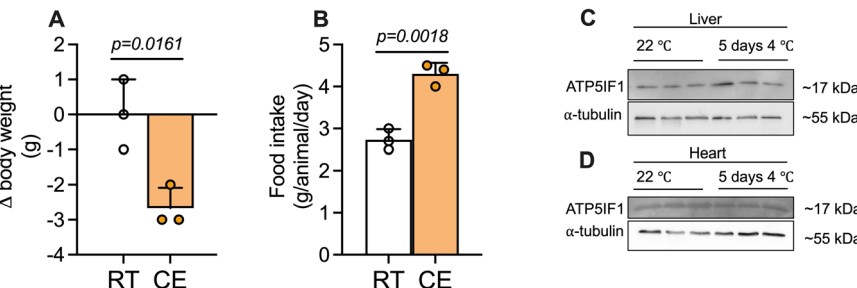

**Figure EV1.  Cold exposure induces body weight loss and does not alter IF1 levels in liver and heart.**

(**A**) Change of body weight and (**B**) food intake during 5 days of cold exposure. IF1 protein levels in (**C**) liver and (**D**) heart after 5 days of cold exposure. IF1 - ATP synthase inhibitory factor subunit 1; Two-tailed Student's t-test (**A**, **B**). Data are expressed with individual values and mean ± SD superimposed. The exact p-value is displayed when $p < 0.05$.

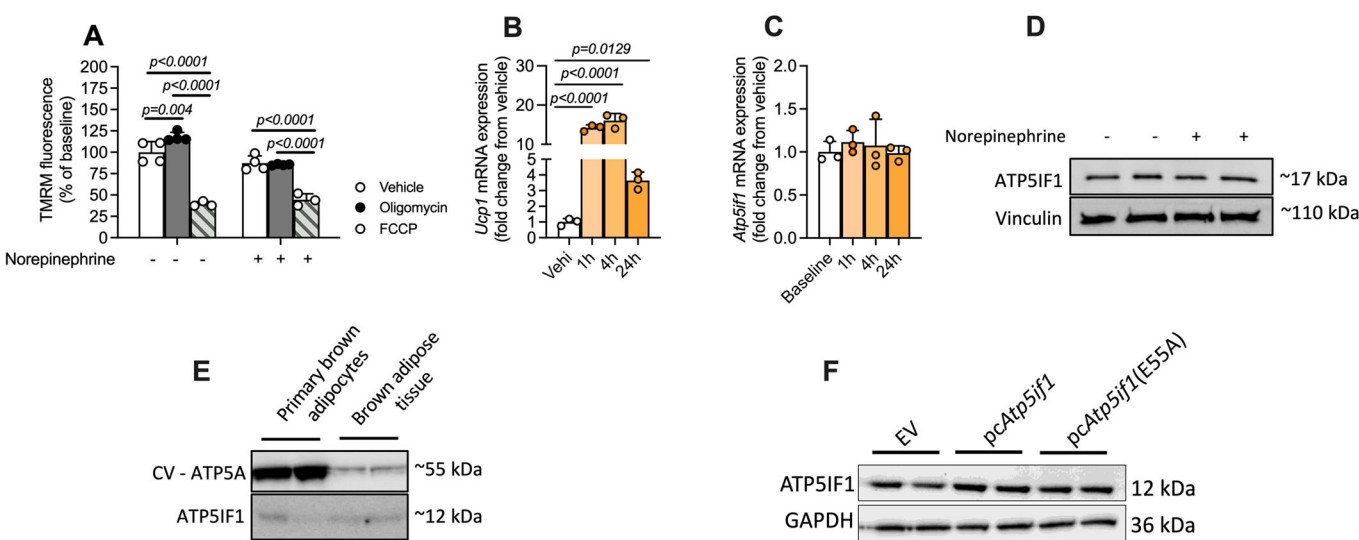

**Figure EV2. Effects of oligomycin and FCCP on mitochondrial membrane potential.**

(A) Brown adipocytes were pre-treated (30 min) with FCCP or oligomycin before the addition of norepinephrine. Norepinephrine treatment lasted 30 min before the cells were loaded with 20 nM TMRM. (B) mRNA levels of *Ucp1* and (C) of *Atp5if1* after 1 h, 4 h, and 24 h of norepinephrine (10 μM) treatment. (D) ATP5IF1 protein levels following norepinephrine stimulation (10 μM for 1 h). (E) Comparison between ATP5A and ATP5IF1 levels in differentiated primary brown adipocytes and brown adipose tissue. (F) Representative blot of IF1 mutant overexpression in differentiated WT1 brown adipocytes. FCCP - Carbonyl cyanide-p-trifluoromethoxyphenylhydrazone. Statistical test: one-way ANOVA followed by LSD post hoc test. Data are expressed with individual values and mean ± SD superimposed. The exact *p*-value is displayed when *p* < 0.05.

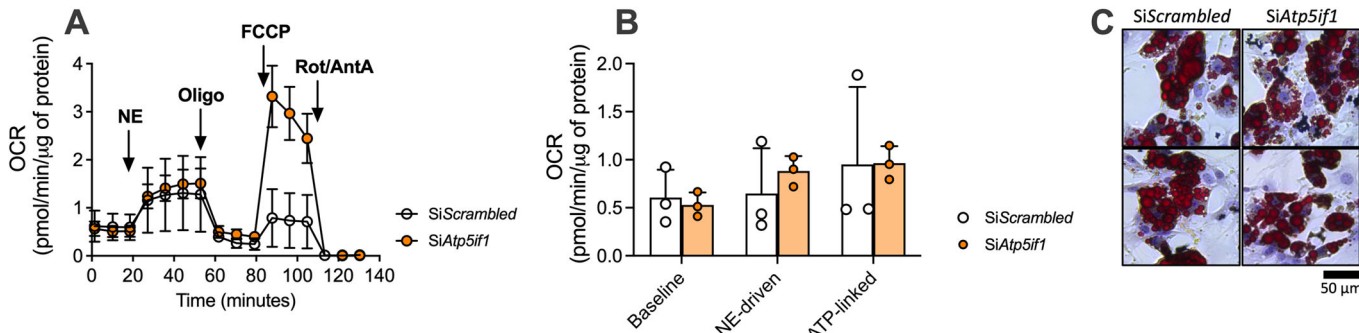

**Figure EV3.  Mitochondrial respiration in the presence of fatty acid-free BSA and lipid accumulation in IF1 knockdown adipocytes.**

(A) Representative trace and (B) quantification of mitochondrial oxygen consumption rate in primary brown adipocytes knockdown for IF1 (si*Atp5if1*) or controls (si*Scrambled*). (C) Lipid content upon *Atp5if1* silencing in primary brown adipocytes visualized by Oil Red O staining. Atp5if1 ATP synthase inhibitory factor subunit 1, OCR oxygen consumption rate, NE norepinephrine, Oligo oligomycin, FCCP carbonyl cyanide-p-trifluoromethoxyphenylhydrazone, Rot rotenone, AA antimycin A. Data are expressed with individual values and mean ± SD superimposed. Two-tailed Student's t-test.

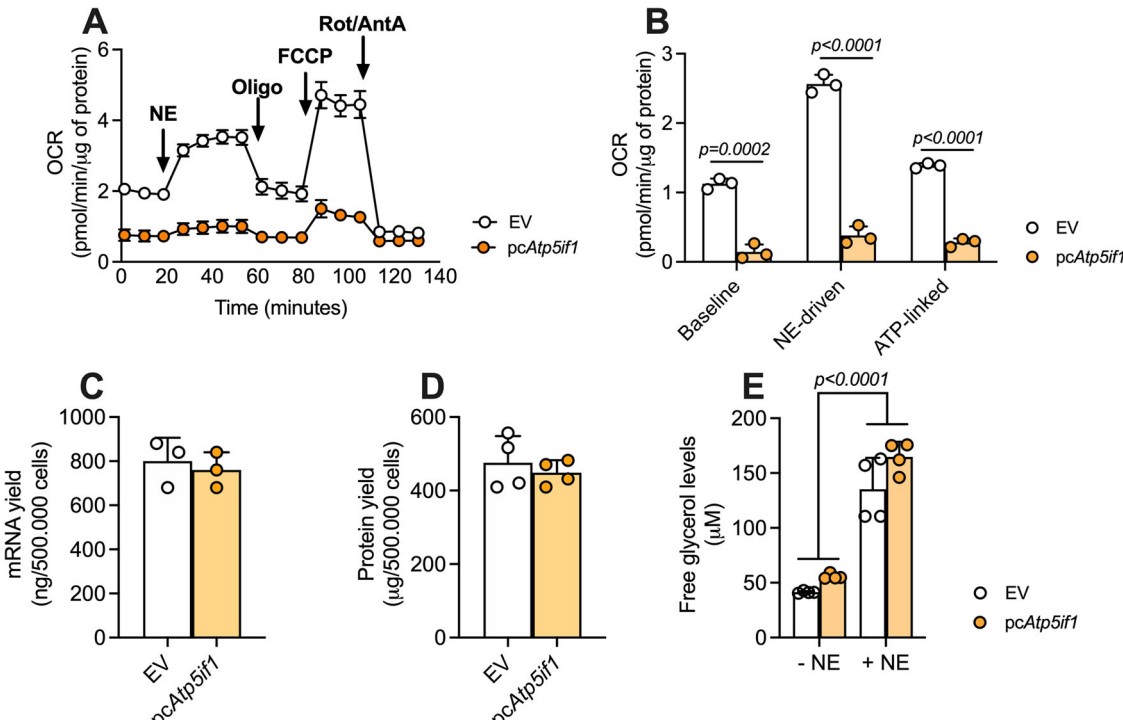

**Figure EV4.  IF1 overexpression suppresses mitochondrial respiration independent of free-fat acids and does not affect basic cell parameters in brown adipocytes.**

(**A**) Representative trace and (**B**) quantification of mitochondrial oxygen consumption rate in primary brown adipocytes overexpressing IF1 (pcAtp5if1) or controls (EV). (**C**) mRNA and (**D**) protein yield from 500,000 cells overexpressing IF1. (**E**) Norepinephrine-induced lipolysis (10 µM for 90 min) in IF1-overexpressing adipocytes. Atp5if1 ATP synthase inhibitory factor subunit 1, OCR oxygen consumption rate, NE norepinephrine, Oligo oligomycin, FCCP carbonyl cyanide-p-trifluoromethoxyphenylhydrazone, Rot rotenone, AA antimycin A. Two-tailed Students t test (**B**, **C**, **D**). Two-way ANOVA followed by LSD post hoc test (**E**). The exact *p*-value is displayed when $p < 0.05$. Data are expressed with individual values and mean ± SD superimposed.

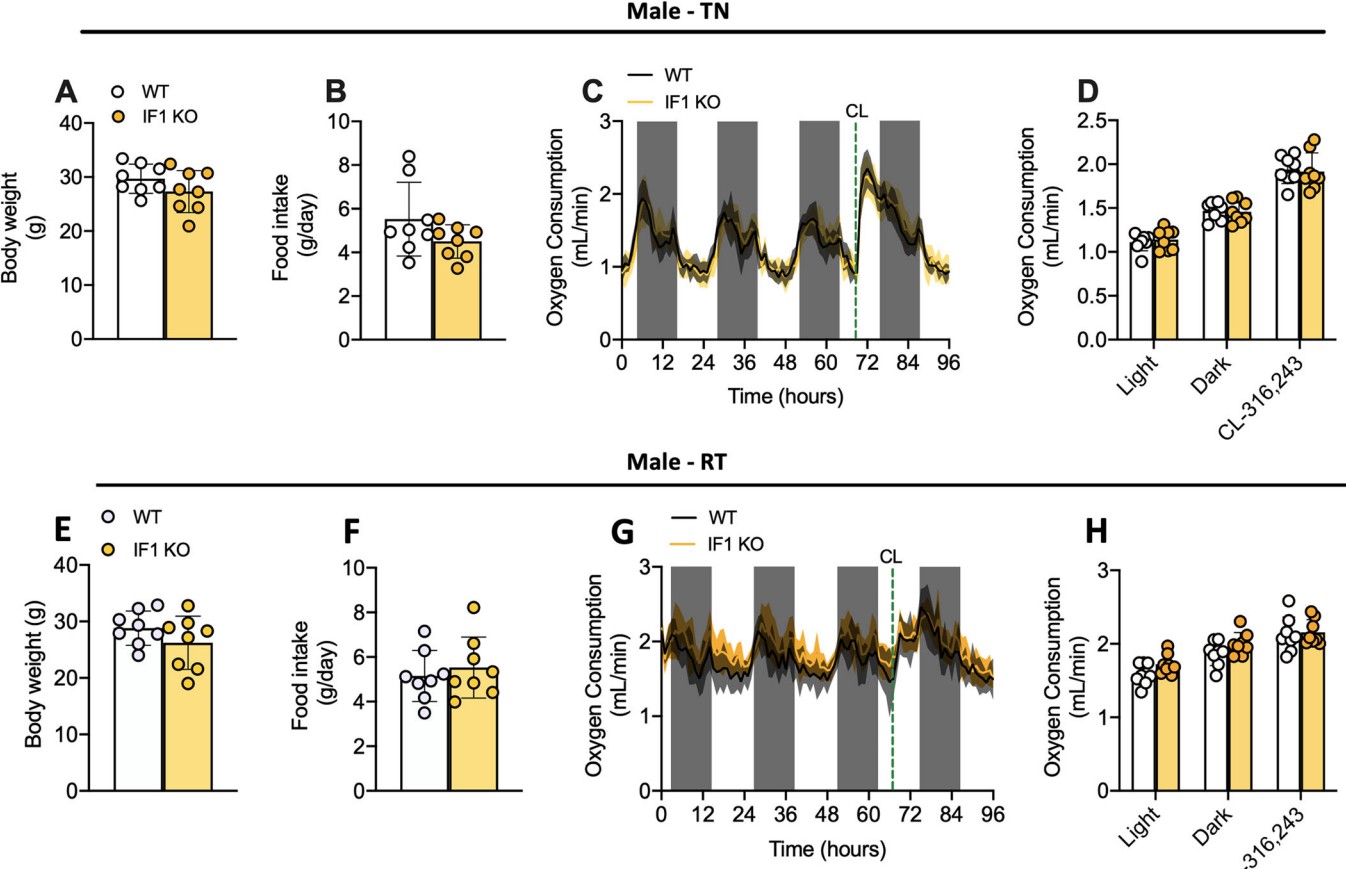

**Figure EV5.** **IF1 global knockout does not affect resting and adrenergic-stimulated whole-body oxygen consumption.**

(A) Body weight, (B) food intake, (C) real-time trace, and (D) average of baseline and CL316,243-induced energy expenditure in adult WT and IF1 KO male mice after 2 weeks of living at thermoneutrality. (E) Body weight, (F) food intake, (G) real-time trace, and (H) average of baseline and CL316,243-induced energy expenditure in adult WT and IF1 KO male mice after 2 weeks living at 23 °C. WT wild type, IF1 KO Mice with global IF1 knockout, RT room temperature, TN thermoneutrality. Two-tailed Student's t-test. Data are expressed with individual values and mean ± SD superimposed.

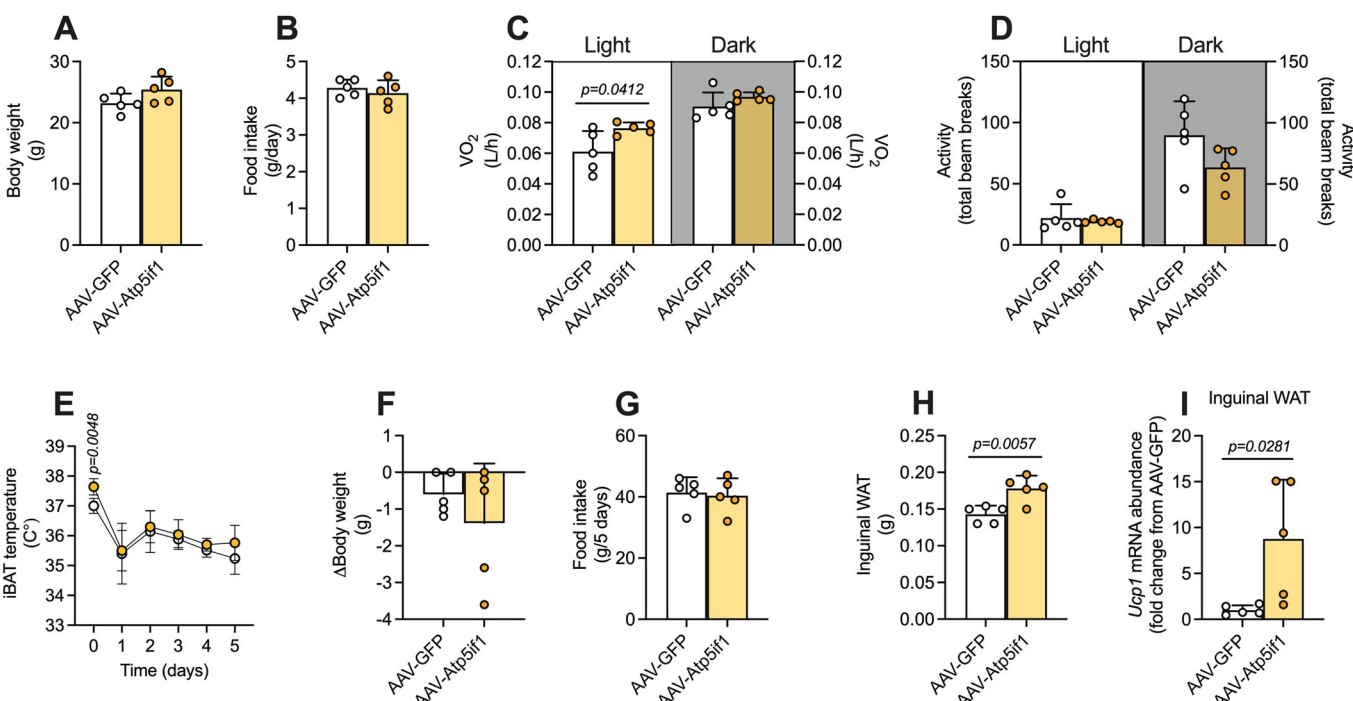

**Figure EV6.  Effects of BAT IF1 overexpression in male mice.**

(**A**) Body weight and (**B**) daily food intake after 14 days of AAV transduction. (**C**) Oxygen consumption and (**D**) voluntary ambulatory activity in the dark and light cycles. (**E**) iBAT temperature at RT (day 0) and following cold exposure (4 °C). (**F**) Change of body weight and (**G**) food intake after 5 days of cold exposure (4 °C). (**H**) Inguinal WAT mass and (**I**) mRNA *Ucp1* levels after 5 days of cold exposure. WAT white adipose tissue. Statistical test: Two-tailed Student's *t*-test. The exact *p*-value is displayed when *p* < 0.05. Data are expressed with individual values and mean ± SD superimposed.

