## [Peer Review File · The EMBO Journal]

IF1 is a cold-regulated switch of ATP synthase hydrolytic activity to support thermogenesis in brown fat

Henver Brunetta, Anna Jung, Fernando Valdivieso-Rivera, Stepheny Zani, Joel Guerra, Vanessa Furino, Annelise Francisco, Marcelo Bercot, Pedro Vieira, Susanne Keipert, Martin Jastroch, Laurent Martinez, Carlos Sponton, Roger Castilho, Marcelo Mori, and Alexander Bartelt

Corresponding authors: Alexander Bartelt (alexander.bartelt@med.uni-muenchen.de) , Marcelo Mori (morima@unicamp.br)

Review Timeline:

Transferred from Review Commons:	30th Apr 24
Editorial Decision:	20th Jun 24
Editor's Correspondence:	30th Jun 24
Revision Received:	3rd Aug 24
Accepted:	16th Aug 24

Editor: Daniel Klimmeck

Transaction Report:

This manuscript was transferred to The EMBO Journal following peer review at Review Commons.

Review #1

1. Evidence, reproducibility and clarity:

Evidence, reproducibility and clarity (Required)

****Summary:**** In the present manuscript, the authors present data in support of their primary discovery that "IF1 controls UCP1-dependent mitochondrial bioenergetics in brown adipocytes". The opening figure convincingly demonstrates that IF1 expression is cold-exposure dependent. They then go on to show that loss of IF1 has functional consequences that would be predicted based on IF1's known role as a regulator of ATP hydrolysis by CV. They go on to make a few additional claims, succinctly detailed in the Discussion section. Specific claims include the following: 1) IF1 is downregulated in cold-adapted BAT, allowing greater hydrolytic activity of ATP synthase by operating in the reverse mode; 2) when IF1 is upregulated in brown adipocytes in vitro mitochondria unable to sustain the MMP upon adrenergic stimulation, 3) IF1 ablation in brown adipocytes phenocopies the metabolic adaptation of BAT to cold, and 4) IF1 overexpression blunts mitochondrial respiration without any apparent compensator response in glycolytic activity. The claims described above are well supported by the evidence. The manuscript is very well written, figures are clear and succinct. Overall, the quality of the work is very high. Given that IF1 is implicated across many fields of study, the novel discovery of IF1 as a regulator of brown adipose mitochondrial bioenergetics will be of significance across several fields. That said, a few areas of concern were apparent. Concerns are detailed in the "Major" and "Minor" comments section below. Additional experiments do not appear to be required, assuming the authors adequately acknowledge the limitations of the study and either remove or qualify speculative claims.

****Major Comments:****

- The authors convincingly demonstrate that IF1 expression is specifically down-regulated in BAT upon cold-exposure. These data strongly implicate a role for IF1 in BAT bioenergetics, a major claim of the authors and a novel finding herein. Additional major strengths of the paper, which provide excellent scientific rigor include the use of both loss of function and gain of function approaches for IF1. In addition, the mutant IF1 experiments are excellent, as they convincingly show that the effects of IF1 are dependent on its ability to bind CV.
- Regarding Figure 1 - Did the content of ATP synthase change? In figure 1A-B, the authors show that ATPase activity of CV is higher in cold-adapted mice. While this result could be due to a loss of IF1, it could also be due to a higher expression of CV. To control for this, the authors should consider blotting for CV, which would allow for ATPase activity to be normalized to expression.
- Regarding MMP generated specifically by ATP hydrolysis at CV, the reversal potential for ANT occurs at a more negative MMP than that of CV (PMID: 21486564). Because reverse transport of ATP (cytosol to matrix) via ANT will also generate a MMP, it is speculative to state that the MMP in the assay is driven by ATP hydrolysis at CV. It is possible and maybe even likely that the majority of the MMP is driven by ANT flux, which in turn limits the amount of ATP hydrolyzed by CV. Admittedly, it is very challenging to differentiate MMP from ANT vs that

from CV, thus the authors simply need to acknowledge that the specific contribution of ATP hydrolysis to MMP remains to be fully determined. That said, the fact that ATP-dependent MMP tracks with IF1 expression does certainly implicate a role for ATP hydrolysis in the process. The authors should consider including a discussion of the ambiguity of the assay to avoid confusion. A role for ANT likely should be incorporated in the Fig. 1J cartoon.

- Regarding the lack of effect of IF1 silencing on MMP, it is possible that IF1 total protein levels are simply lower in cultured brown fat cells relative to tissue? The authors could consider testing this by blotting for IF1 and CV in BAT and brown fat cells. The ratio of IF1/ATP5A1 in tissue versus cells may provide some amount of mechanistic evidence as to their findings.

- The calculation of ATP synthesis from respiration sensitive to oligomycin has many conceptual flaws. Unlike glycolysis, where ATP is produced via substrate level phosphorylation, during OXPHOS, the stoichiometry of ATP produced per 2e transfer is not known in intact brown adipose cells. This is a major limitation of this "calculated ATP synthesis" approach that is beginning to become common. Such claims are speculative and thus likely do more harm than good. In addition to ANT and CV, there are many proton-consuming reactions driven by the proton motive force (e.g., metabolite transport, Ca²⁺ cycling, NADPH synthesis). Although it remains unclear how much proton conductance is diverted to non ATP synthesis dependent processes, it seems highly likely that these processes contribute to respiratory demand inside living cells. Moreover, just as occurs with UCP1 in response to adrenergic stimuli, proton conductance across the various proton-dependent processes likely changes depending on the cellular context, which is another reason why using a fixed stoichiometry to calculate how much ATP is produced from oxygen consumption is so highly flawed. Maximal P/O values that are often used for NAD/FAD linked flux are generated using experimental conditions that favor near complete flux through the ATP synthesis system (supraphysiological substrate and ADP levels). The true P/O value inside living cells is likely to be lower.

- Why are the results in Figure 3K expressed as a % of basal? Could the authors please normalize the OCR data to protein and/or provide a justification for why different normalization strategies were used between 3K and 3M?

- The authors claim that IF1 overexpression lowers ATP production via OXPHOS. However, given the major limitations of this assay (as discussed above), these claims should be viewed as speculation. This needs to be addressed by the authors as a major limitation. The fact that the ATP/ADP levels did not change do not support a reduction in ATP production, as claimed in the title of Figure 4.

- In the discussion, the authors state "However, considering that IF1 inhibits F₁-ATP synthase in a 1:1 stoichiometric ratio, the relatively higher expression of IF1 in BAT at room temperature could represent an additional inhibitory factor for ATP synthesis in this tissue." This does not appear to be correct. Although IF1 has been suggested to partially lower maximal rates of ATP synthesis rates, most of this evidence comes from over-expression experiments. According to the current understanding of IF1-CV interaction, the protein is expelled from the complex during rotation in favor of ATP synthesis (PMID: 37002198). It is far more likely that ATP synthesis is low in BAT mitochondria due to the low CV expression. Relative to heart and when normalized to mitochondrial content, CV expression in BAT mitochondria is about 10% that of heart (PMID: 33077793).

- The last sentence of the manuscript states, "Given the importance of IF1 to control brown adipocyte energy metabolism, lowering IF1 levels therapeutically might enhance approaches to enhance NST for improving cardiometabolic health in humans." This sentence seems at odds

with the evidence that IF1 levels go up, not down, in human BAT upon cold exposure.

****Minor Comments:****

- The term "anaerobic glycolysis" is used throughout. All experiments were performed under normoxic conditions, thus the correct term is "aerobic glycolysis."
- Only male mice were used in the study, could the authors please provide a justification for this?

2. Significance:

Significance (Required)

Overall, the quality of the work is very high. Given that IF1 is implicated across many fields of study, the novel discovery of IF1 as a regulator of brown adipose mitochondrial bioenergetics will be of significance across several fields.

My expertise is in mitochondrial thermodynamics; thus, I do not feel there are any parts of the paper that I do not have sufficient expertise to evaluate.

3. How much time do you estimate the authors will need to complete the suggested revisions:

Estimated time to Complete Revisions (Required)

(Decision Recommendation)

Less than 1 month

Yes

Review #2

1. Evidence, reproducibility and clarity:

Evidence, reproducibility and clarity (Required)

****Summary****

The manuscript by Brunetta and colleagues conveys the message that the ATPase inhibitory factor 1 (IF1) protein, a physiological inhibitor of mitochondrial ATP synthase, is expressed in BAT of C57BL/6J mice. Moreover, upon cold-adaptation of mice they report that the content of IF1 in BAT is downregulated to sustain the mitochondrial membrane potential (MMP) as a result of reverse functioning of the enzyme. In experiments of loss and gain of function of IF1 in cultured brown adipocytes and WT cells they further stress that IF1 silencing promotes metabolic reprogramming to an enhanced glycolysis and lipid oxidation, whereas IF1 overexpression blunts ATP production rendering a quiescent cellular state of the adipocytes.

2. Significance:

Significance (Required)

Claims and conclusions:

I have been surprised by the claim that IF1 protein is expressed in BAT under basal conditions and that its expression is downregulated in the cold-adapted tissue. In a previously published work by Forner et al., (2009) *Cell Metab* 10, 324-335 (reference 43), using a quantitative proteomic approach, it is reported that the mitochondrial proteome of mouse BAT under basal conditions contains a low content of IF1 (at level comparable to the background of the analysis). Remarkably, in the same study they show that there is roughly a 2-fold increase in the content of IF1 protein in mitochondria of BAT at 4d and 24d of cold-adaptation of mice. In other words, just the opposite of what is being reported in the Brunetta study. Importantly, the last paragraph of the discussion needs to be amended when mentioning the work of Forner et al. (ref.43). The mentioned reference studied changes in the mouse mitochondrial proteome not in human mitochondria, as it is stated in the alluded paragraph.

More puzzling are the western blots in Figures 1E, 1H, Supp. Fig. 1C, D where IF1 (ATP5IF1) is identified by a 17kDa band. However, in other Figures (Fig. 2, Fig. 3, Fig. 4, Supp Fig. 2) IF1 is identified by its well-known 12kDa band. What is the reason for this change in labeling of the IF1 band? The reactivity of the anti-IF1 antibody used? It has been previously documented that liver of C57BL/6J and FVB mouse strains do not express IF1 to a significant level when compared to heart IF1 levels (Esparza-Molto (2019) *FASEB J.* 33, 1836-1851). However, in Fig. 1E they show opposite findings, much higher levels of IF1 in liver than in heart as revealed by the 17kDa band. Moreover, in Fig. 1H they show the vanishing of the 17 kDa band under cold adaptation, which is not the migration of IF1 in gels as shown in their own figures (see Fig. 2,

Fig. 3, Fig. 4, Supp Fig. 2). I am certainly reluctant to accept that the 17kDa band shown in Figures 1E, 1H, Supp. Fig. 1C, D is indeed IF1. Most likely it represents a non-specific protein recognized by the antibody in the tissue extracts analyzed. Cellular overexpression experiments of IF1 in WT1 cells (Fig. 2E) and primary brown adipocytes (Fig. 4B) also support this argument.

Overall, I do not support publication of this study for the reasons stated above.

3. How much time do you estimate the authors will need to complete the suggested revisions:

Estimated time to Complete Revisions (Required)

(Decision Recommendation)

More than 6 months

Yes

Review #3

1. Evidence, reproducibility and clarity:

Evidence, reproducibility and clarity (Required)

****Summary:****

In this manuscript, Brunnetta et al describe the role of IF1 in brown adipose tissue activation using in vivo and in vitro experimental models. They observed that cold adaptation promotes a reduction in IF1 expression and an increase in the reverse activity of mitochondrial ATPase or

Complex V. Based on these results, the authors explore the contribution of IF1 in this metabolic pathway by modeling the thermogenic process in differentiated primary brown adipocytes. They silenced and overexpressed IF1 in culture and studied their adrenergic stimulation under norepinephrine.

****Major comments:****

The experiments are well explained and the manuscript flows very well. There are several comments that should be addressed.

1. The authors measure ATP hydrolysis in isolated mitochondria from BAT in Figure 1. They observed that IF1 is decreased upon cold exposure and that ATP hydrolysis is increased. They assess protein levels of different OXPHOS proteins, including IF1 but not other proteins of Complex V (ATP5A) as they do in Figures 3 and 4. It is important to see that cold exposure only affects IF1 levels but not other proteins from Complex V. Does IF1/Complex V ratio change?
2. In Figure 2J, the drop in MMP is lower upon adrenergic stimulation than in Figure 2E. The same observation applies to other results when the reduction in MMP after NE addition is minimal. Why do the authors remove TMRM for the measurements of membrane potential? TMRM imaging is normally done in the presence of the dye in non-quenching mode. Treatments should be done prior to the addition of the dye and then TMRM should be added and left during the imaging analysis and measure in non-quenching mode. This might explain some of the above-mentioned points regarding the MMP data. Alternatively, if the dye is removed before the measurements, they should let the cells to adapt and so the dye equilibrates between mitochondria and cytosol. A more elegant method to measure membrane potential could be live-cell imaging. In addition, authors propose that mitochondrial membrane potential upon NE stimulation is maintained by reversal of ATP synthase. If this is the case, one would expect that addition of oligomycin in NE treated adipocytes would cause depolarization. However, in FigS2A this is not the case. Authors should comment on this in addition to considering more elegant approach to measure MMP
3. In Figure 2, in the model of siIF1, there is baseline more phosphorylation of AMPK than in the scramble control (pAMPK). However, this is not the case of p-p38MAPK. Do the authors have any explanation for those differences in baseline activation of the stress kinases when IF1 is silenced? In the same experimental group, addition of NE seems to have more effect in the scrambled than in siIF1, but the plotted data does not reflect these differences. In contrast, increase in pAMPK upon NE is higher in IF1 overexpressing cells compared to EV (Figure 2H), but again this is not reflected in western blot quantification (Figure 2I).
4. Does NE promote decrease of IF1 expression in control (siScramble and EV) adipocytes? The authors should test it and see whether it goes in the same direction as the observations derived from the experiments in cold exposed mice. This is very important point, as it could explain the lack of an additional effect of IF1 silencing on NE-induced depolarization (Figure 2E).
5. Does NE promote decrease of IF1 expression in the scramble and EV adipocytes? The authors should test it and see whether it goes in the same direction as the observations derived from the experiments in cold exposed mice.
6. For MMP data in Fig2, they should include significance between non treated and NE-treated groups. They say: "While UCP1 ablation did not cause any effect on MMP upon adrenergic stimulation...", but NE caused (probably significant) depolarization in siUCP1, which seems

even stronger than depolarization in EV. This is opposite to what you would expect. They also didn't confirm UCP1 silencing with western blot.

7. It has been established that decreased expression of IF1 promotes increase in the reverse activity of Complex V, ATP hydrolytic activity. Increase in ATP hydrolysis also affects ECAR. The authors should consider this when calculating the contribution of ATP glycolysis versus ATP OXPHOS since the ATP hydrolysis is also playing a role in the ECAR increase. The data should be reinterpreted. ATP hydrolysis should be measured in the situation where IF1 is silenced and overexpressed. These measurements can be done in cells using the Seahorse.

8. The authors use GAPDH as loading control in western blots. They should use another protein since GAPDH is part of the intermediary metabolism and plays a role in glycolysis.

9. The authors show that reduction of IF1 involves more lipid utilization. They should include more experiments showing the connection of the metabolic adaptation in the absence of IF1 and some lipid imaging.

10. In the text, "Despite this adjustment of experimental conditions, we did not detect any effect of IF1 ablation on mitochondrial oxygen consumption (Supplementary Fig. 3A,B)", this is true for baseline, NE-driven and ATP-linked respiration, but what about maximal respiration? There is a huge increase in IF1 knockdown... They should explain these results.

11. In Figure 3K they present OCR as % of baseline, but in a similar experiment in Figure 4G it is OCR/protein, they should make the Y axis consistent across experiments.

12. The graphical abstract is confusing. In BAT there are two populations of mitochondria, the cytosolic and the mitochondria attached to the lipid droplet, peridroplet mitochondria (PDM). Upon adrenergic stimulation, PDM leave the lipid droplet and lipolysis takes place. The authors propose that upon adrenergic stimulation, IF1 is reduced and there is lipid mobilization. The part of the scheme where it says "fully recruited" should be removed or rewritten, since adrenergic stimulation is not compatible with mitochondria recruitment around the lipid droplet.

13. The title should be rewritten to better reflect the research presented in the manuscript.

****Minor comments:****

1. Some of the Y axis should be corrected. For example, in Figure 2J, L and M should say % of EV untreated, Similarly, in Figure 2E, it should say % of scramble untreated. In Figure 3N, the Y axis is misspelled. All the Y axis referring to percentages should have the same scale for comparison purposes.

2. The authors should describe better the results corresponding to Figure 2. There is a lot of information and they should improve the description pertaining the connection between the different pieces of data relating the different signaling pathways that are shown. For westerns in this Figure, they should provide some rationale (one to two sentences in the results section) as to why they are checking the expression of pAMPK and p38-MAPK.

****Here are some comments referring to the methods section:****

1. For Complex V hydrolytic activity, the reaction buffer contains 10mM Na-azide. I guess this is to inhibit respiration, but wouldn't azide also inhibit complex V at this concentration?

2. In the mitochondrial isolation protocol, they say "mitochondria were centrifuged at 800g for

10min..." Will this speed pellet the mitochondria? I think this is a mistake in writing.

3. For the safranin-O experiment, they don't mention mitochondrial substrate used, probably it's in the reference that they provide, but I think it should be included in the text.

2. Significance:

Significance (Required)

The manuscript is well written, and it flows well when reading. However, there are some additional experiments that need to be performed to reach the conclusions the authors claim.

The role of ATP hydrolysis in BAT thermogenesis is novel and interesting as it can shed some light onto potential approaches to promote BAT activation.

3. How much time do you estimate the authors will need to complete the suggested revisions:

Estimated time to Complete Revisions (Required)

(Decision Recommendation)

Between 1 and 3 months

Yes

Review #4

1. Evidence, reproducibility and clarity:

Evidence, reproducibility and clarity (Required)

This is an interesting investigation into the activity of IF1 in brown adipocytes. The findings are innovative and the conclusion is well-supported by the data. The conclusion is in line with previous reports on IF1 activities in other cell types, particularly in terms of its regulation of FoF1-ATPase. The authors have executed an exceptional job in designing the study, preparing the figures, and writing the manuscript. Overall, this study significantly contributes to the understanding of IF1 activity in brown adipocytes and its role in thermogenesis.

2. Significance:

Significance (Required)

The study demonstrates involvement of IF1 in regulating thermogenesis in brown adipocytes, which is a unique aspect not covered in existing literature. Advantage of the study is well-designed cellular studies. The major weakness is lack of proof of conclusion in vivo. There are a few minor concerns that should be addressed to further enhance quality of the manuscript.

1. Current discussion does not mention the regulation of IF1 protein by the cAMP/PKA pathway. This point should be included to provide a comprehensive understanding of the regulatory mechanisms of IF1 protein.
2. It has been reported that IF1 also influences the structure of mitochondrial crista. Considering the observed changes with IF1 knockdown, it would be valuable to discuss this activity in relation to the findings of the study.

3. How much time do you estimate the authors will need to complete the suggested revisions:

Estimated time to Complete Revisions (Required)

(Decision Recommendation)

Less than 1 month

Yes

RC-2023-02105R: Brunetta et al.,

IF1 is a cold-regulated switch of ATP synthase to support thermogenesis in brown fat

We are happy to submit our revised manuscript after considering the suggestions made by reviewers. The comments were overall positive, and the changes requested were mostly editorial. We have, nevertheless, added new experiments as quality controls. These experiments did not affect the main conclusions of our work. In addition, we also included two *in vivo* experimental models of gain and loss-of-function, to further address the physiological relevance of IF1 in BAT thermogenesis. We believe with these additional experiments, quality controls as well as *in vivo* models, our study has improved considerably. We hope our efforts will be appreciated by the reviewers and we make ourselves available to answer any further questions.

Reviewer #1 (Evidence, reproducibility and clarity (Required)):

Summary: In the present manuscript, the authors present data in support of their primary discovery that "IF1 controls UCP1-dependent mitochondrial bioenergetics in brown adipocytes". The opening figure convincingly demonstrates that IF1 expression is cold-exposure dependent. They then go on to show that loss of IF1 has functional consequences that would be predicted based on IF1's known role as a regulator of ATP hydrolysis by CV. They go on to make a few additional claims, succinctly detailed in the Discussion section. Specific claims include the following: 1) IF1 is downregulated in cold-adapted BAT, allowing greater hydrolytic activity of ATP synthase by operating in the reverse mode; 2) when IF1 is upregulated in brown adipocytes *in vitro* mitochondria unable to sustain the MMP upon adrenergic stimulation, 3) IF1 ablation in brown adipocytes phenocopies the metabolic adaptation of BAT to cold, and 4) IF1 overexpression blunts mitochondrial respiration without any apparent compensator response in glycolytic activity. The claims described above are well supported by the evidence. The manuscript is very well written, figures are clear and succinct. Overall, the quality of the work is very high. Given that IF1 is implicated across many fields of study, the novel discovery of IF1 as a regulator of brown adipose mitochondrial bioenergetics will be of significance across several fields. That said, a few areas of concern were apparent. Concerns are detailed in the "Major" and "Minor" comments section below. Additional experiments do not appear to be required, assuming the authors adequately acknowledge the limitations of the study and either remove or qualify speculative claims.

Major Comments:

1. The authors convincingly demonstrate that IF1 expression is specifically down-regulated in BAT upon cold-exposure. These data strongly implicate a role for IF1 in BAT bioenergetics, a major claim of the authors and a novel finding herein. Additional major strengths of the paper, which provide excellent scientific rigor include the use of both loss of function and gain of function approaches for IF1. In addition, the mutant IF1 experiments are excellent, as they convincingly show that the effects of IF1 are dependent on its ability to bind CV.

RESPONSE: We thank the reviewer for the positive feedback on our work.

2. Regarding Figure 1 - Did the content of ATP synthase change? In figure 1A-B, the authors show that ATPase activity of CV is higher in cold-adapted mice. While this result could be due to a loss of IF1, it could also be due to a higher expression of CV. To control for this, the authors should consider blotting for CV, which would allow for ATPase activity to be normalized to expression.

RESPONSE: Thank you for this suggestion. We have now determined complex V subunit A in our experimental protocol. We found that cold exposure does not impact complex V protein levels. Given the importance of this control, we have now included it in Figure 1 (Please, see the revised version) alongside the IF1/complex V ratio. In addition, we have now performed WBs in the BAT from mice exposed for 3 and 7 days to thermoneutrality (~28°C). We found that IF1 is not reduced following whitening of BAT by this approach whilst UCP1 and other mitochondrial proteins are reduced. This set of data is now included in Figure 1I,K,L.

3. Regarding MMP generated specifically by ATP hydrolysis at CV, the reversal potential for ANT occurs at a more negative MMP than that of CV (PMID: 21486564). Because reverse transport of ATP (cytosol to matrix) via ANT will also generate a MMP, it is speculative to state that the MMP in the assay is driven by ATP hydrolysis at CV. It is possible and maybe even likely that the majority of the MMP is driven by ANT flux, which in turn limits the amount of ATP hydrolyzed by CV. Admittedly, it is very challenging to different MMP from ANT vs that from CV, thus the authors simply need to acknowledge that the specific contribution of ATP hydrolysis to MMP remains to be fully determined. That said, the fact that ATP-dependent MMP tracks with IF1 expression does certainly implicate a role for ATP hydrolysis in the process. The authors should consider including a discussion of the ambiguity of the assay to avoid confusion. A role for ANT likely should be incorporated in the Fig. 1J cartoon.

RESPONSE: Thank you for bringing the ANT contribution to MMP to our attention. The effects of ATP in the real-time MMP measurements were totally abolished by the addition of oligomycin in BAT-derived isolated mitochondria, thus suggesting dependency of complex V in this process. However, the assessment of MMP in intact cells is much more challenging given cytosolic vs. mitochondrial contribution to ATP pool, and ATP synthase vs. ANT reversal capacity depending on MMP. Nevertheless, we have addressed these points in the discussion section as well as added to our schematic cartoon in Figure 1m.

4. Regarding the lack of effect of IF1 silencing on MMP, it is possible that IF1 total protein levels are simply lower in cultured brown fat cells relative to tissue? The authors could consider testing this by blotting for IF1 and CV in BAT and brown fat cells. The ratio of IF1/ATP5A1 in tissue versus cells may provide some amount of mechanistic evidence as to their findings.

RESPONSE: We have now blotted for complex V and IF1 in both differentiated primary brown adipocytes and BAT homogenates derived from mice kept at room temperature (~22°C). We found the levels of complex V in primary brown adipocytes are higher than BAT homogenates. Therefore, IF1/complex V ratio is different between these two systems. This has indeed the potential to influence our gain and loss-of-function experiments. We have added these results alongside their interpretation in the revised manuscript.

5. The calculation of ATP synthesis from respiration sensitive to oligomycin has many conceptual flaws. Unlike glycolysis, where ATP is produced via substrate level phosphorylation, during OXPHOS, the stoichiometry of ATP produced per 2e transfer is not known in intact brown adipose cells. This is a major limitation of this "calculated ATP synthesis"

approach that is beginning to become common. Such claims are speculative and thus likely do more harm than good. In addition to ANT and CV, there are many proton-consuming reactions driven by the proton motive force (e.g., metabolite transport, Ca²⁺ cycling, NADPH synthesis). Although it remains unclear how much proton conductance is diverted to non-ATP synthesis dependent processes, it seems highly likely that these processes contribute to respiratory demand inside living cells. Moreover, just as occurs with UCP1 in response to adrenergic stimuli, proton conductance across the various proton-dependent processes likely changes depending on the cellular context, which is another reason why using a fixed stoichiometry to calculate how much ATP is produced from oxygen consumption is so highly flawed. Maximal P/O values that are often used for NAD/FAD linked flux are generated using experimental conditions that favor near complete flux through the ATP synthesis system (supraphysiological substrate and ADP levels). The true P/O value inside living cells is likely to be lower.

RESPONSE: We agree with the reviewer regarding the limitations on calculating ATP production in intact cells based on respiration and proton flux. However, this was only one experiment on which we based our conclusions, as these were also supported by i.e. ATP/ADP ratio measurements and oxygen consumption using different substrates. Therefore, we do not rely exclusively on the ATP production estimative, rather we use this experiment to support complementary methodologies. Nevertheless, we have now better detailed our experimental protocol as well as acknowledged the limitations of the method, so the reader is aware of our procedure and its limitations. We hope the reviewer understands our motivation to perform these experiments and the contribution to our study.

6. Why are the results in Figure 3K expressed as a % of basal? Could the authors please normalize the OCR data to protein and/or provide a justification for why different normalization strategies were used between 3K and 3M?

RESPONSE: We apologize for the lack of consistency. We have now updated Figure 3 to show all the data in absolute values divided by protein content. This change does not affect the overall interpretation of the findings.

7. The authors claim that IF1 overexpression lowers ATP production via OXPHOS. However, given the major limitations of this assay (as discussed above), these claims should be viewed as speculation. This needs to be addressed by the authors as a major limitation. The fact that the ATP/ADP levels did not change do not support a reduction in ATP production, as claimed in the title of Figure 4.

RESPONSE: The reduction in ATP levels and mitochondrial respiration (independent of the substrate offered) suggests a reduction in ATP production rather than an increase in ATP consumption. Moreover, the maintenance of ATP/ADP ratio suggests the existence of a compensatory mechanism to avoid cellular energy crises, which we interpreted as reduced metabolic activity of the cells. Nevertheless, we have now reworded our statements to address the limitations of the methods and our interpretation of the data.

8. In the discussion, the authors state "However, considering that IF1 inhibits F1-ATP synthase in a 1:1 stoichiometric ratio, the relatively higher expression of IF1 in BAT at room temperature could represent an additional inhibitory factor for ATP synthesis in this tissue." This does not appear to be correct. Although IF1 has been suggested to partially lower maximal rates of ATP synthesis rates, most of this evidence comes from over-expression experiments. According to the current understanding of IF1-CV interaction, the protein is expelled from the complex

during rotation in favor of ATP synthesis (PMID: 37002198). It is far more likely that ATP synthesis is low in BAT mitochondria due to the low CV expression. Relative to heart and when normalized to mitochondrial content, CV expression in BAT mitochondria is about 10% that of heart (PMID: 33077793).

RESPONSE: We agree with the reviewer and removed this sentence.

9. The last sentence of the manuscript states, "Given the importance of IF1 to control brown adipocyte energy metabolism, lowering IF1 levels therapeutically might enhance approaches to enhance NST for improving cardiometabolic health in humans." This sentence seems at odds with the evidence that IF1 levels go up, not down, in human BAT upon cold exposure.

RESPONSE: In light of our new experiments, we have now updated our conclusions.

Minor Comments:

10. The term "anaerobic glycolysis" is used throughout. All experiments were performed under normoxic conditions, thus the correct term is "aerobic glycolysis."

RESPONSE: Thank you for this comment and we have replaced this term as suggested.

11. Only male mice were used in the study, could the authors please provide a justification for this?

RESPONSE: Given we devoted most of our efforts to the manipulation of IF1 *in vitro*, we have used the mouse model as a proof-of-principle on the impact of IF1 in adrenergic-induced thermogenesis. We have now included IF1 KO male and female mice to address the role of IF1 in adrenergic-induced thermogenesis. However, due to the limitation of material, we could only perform AAV *in vivo* gain-of-function in male mice, therefore, our results cannot be immediately transferred to both sexes, unfortunately.

Reviewer #1 (Significance (Required)):

Overall, the quality of the work is very high. Given that IF1 is implicated across many fields of study, the novel discovery of IF1 as a regulator of brown adipose mitochondrial bioenergetics will be of significance across several fields.

My expertise is in mitochondrial thermodynamics; thus, I do not feel there are any parts of the paper that I do not have sufficient expertise to evaluate.

Reviewer #2 (Evidence, reproducibility and clarity (Required)):

Summary

The manuscript by Brunetta and colleagues conveys the message that the ATPase inhibitory factor 1 (IF1) protein, a physiological inhibitor of mitochondrial ATP synthase, is expressed in BAT of C57BL/6J mice. Moreover, upon cold-adaptation of mice they report that the content of IF1 in BAT is downregulated to sustain the mitochondrial membrane potential (MMP) as a result of reverse functioning of the enzyme. In experiments of loss and gain of function of IF1 in cultured brown adipocytes and WT cells they further stress that IF1 silencing promotes metabolic reprogramming to an enhanced glycolysis and lipid oxidation, whereas IF1 overexpression blunts ATP production rendering a quiescent cellular state of the adipocytes.

RESPONSE: We appreciate the time the reviewer invested in our work. Please, see our responses below in a point-by-point manner.

Reviewer #2 (Significance (Required)):

Claims and conclusions:

I have been surprised by the claim that IF1 protein is expressed in BAT under basal conditions and that its expression is downregulated in the cold-adapted tissue. In a previously published work by Forner et al., (2009) Cell Metab 10, 324-335 (reference 43), using a quantitative proteomic approach, it is reported that the mitochondrial proteome of mouse BAT under basal conditions contains a low content of IF1 (at level comparable to the background of the analysis). Remarkably, in the same study they show that there is roughly a 2-fold increase in the content of IF1 protein in mitochondria of BAT at 4d and 24d of cold-adaptation of mice. In other words, just the opposite of what is being reported in the Brunetta study.

RESPONSE: We are aware of the inconsistencies between our findings and Forner et al. (2009). We would like to point out that we have determined IF1 levels in BAT in two separate cohorts with the same findings, and in a third cohort, we observed IF1 mRNA levels to be downregulated in a much shorter timeframe. Our functional analysis is in line with this pattern of regulation. A closer look at the supplementary table provided by Forner et al. (2009), shows that the increase in IF1 content following cold exposure is not supported and since we do not have further insight into the methods and analysis employed by the Forner et al. group, we believe a direct comparison should be avoided at the moment. Regarding the baseline levels of IF1 in BAT, the relatively high abundance of IF1 in BAT was also found by another independent group (<https://doi.org/10.1101/2020.09.24.311076>).

Importantly, the last paragraph of the discussion needs to be amended when mentioning the work of Forner et al. (ref.43). The mentioned reference studied changes in the mouse mitochondrial proteome not in human mitochondria, as it is stated in the alluded paragraph.

RESPONSE: We apologize for this oversight; we have now reworded our statement.

More puzzling are the western blots in Figures 1E, 1H, Supp. Fig. 1C, D where IF1 (ATP5IF1) is identified by a 17kDa band. However, in other Figures (Fig. 2, Fig. 3, Fig. 4, Supp Fig. 2) IF1 is identified by its well-known 12kDa band. What is the reason for this change in labeling of the IF1 band? The reactivity of the anti-IF1 antibody used? It has been previously documented that liver of C57BL/6J and FVB mouse strains do not express IF1 to a significant level when compared to heart IF1 levels (Esparza-Molto (2019) FASEB J. 33, 1836-1851). However, in Fig. 1E they show opposite findings, much higher levels of IF1 in liver than in

heart as reveal by the 17kDa band. Moreover, in Fig. 1H they show the vanishing of the 17 kDa band under cold adaptation, which is not the migration of IF1 in gels as shown in their own figures (see Fig. 2, Fig. 3, Fig. 4, Supp Fig. 2). I am certainly reluctant to accept that the 17kDa band shown in Figures 1E, 1H, Supp. Fig. 1C, D is indeed IF1. Most likely it represents a non-specific protein recognized by the antibody in the tissue extracts analyzed. Cellular overexpression experiments of IF1 in WT1 cells (Fig. 2E) and primary brown adipocytes (Fig. 4B) also support this argument. Overall, I do not support publication of this study for the reasons stated above.

RESPONSE: We understand the concerns raised by the reviewer and apologize for the lack of details in our experimental procedures. While we used the same antibody in the study (Cell Sig. cat. Num. 8528, 1:500), we used two different types of gels. The difference in the molecular weight appearance of IF1 is likely through the migration of the protein in the agarose gel. By using custom-made gels, we observe the protein ~17kDa (Fig. 1 and 5), whereas by using commercial gels (Fig. 2, 3, and 4), we observe the protein closer to the predicted molecular weight (i.e. ~12kDa). Of note, gain and loss-of-function experiments, both *in vivo* as well as *in vitro* confirm this statement and the specificity of the antibody (Fig. 2, 3, 4, 5, Fig. EV2). In addition, when we ran a custom-made gel with primary BAT cells, we observed again the ~17kDa band (see Figure for the reviewer below). These experiments alongside the absence of other bands in the gels (see uncropped membranes in Supplementary Figure 1) make us conclude that the band we observe is indeed IF1. Nevertheless, we have now updated our methods section, so the reader is aware of our approaches. We hope the reviewer is satisfied with our additional experiments and editions throughout the manuscript.

Reviewer #3 (Evidence, reproducibility and clarity (Required)):

Summary:

In this manuscript, Brunnetta et al describe the role of IF1 in brown adipose tissue activation using in vivo and in vitro experimental models. They observed that cold adaptation promotes a reduction in IF1 expression and an increase in the reverse activity of mitochondrial ATPase or Complex V. Based on these results, the authors explore the contribution of IF1 in this metabolic pathway by modeling the thermogenic process in differentiated primary brown adipocytes. They silenced and overexpressed IF1 in culture and studied their adrenergic stimulation under norepinephrine.

Major comments:

The experiments are well explained and the manuscript flows very well. There are several comments that should be addressed.

RESPONSE: We thank the reviewer for the kind words regarding our work.

1. The authors measure ATP hydrolysis in isolated mitochondria from BAT in Figure 1. They observed that IF1 is decreased upon cold exposure and that ATP hydrolysis is increased. They assess protein levels of different OXPHOS proteins, including IF1 but not other proteins of Complex V (ATP5A) as they do in Figures 3 and 4. It is important to see that cold exposure only affects IF1 levels but not other proteins from Complex V. Does IF1/Complex V ratio change?

RESPONSE: We thank the reviewer for this suggestion which was also raised by Reviewer #1. We have now measured complex V subunit A in our experimental protocol. We found that cold exposure does not impact complex V protein levels. Given the importance of this information, we have now included it in Figure 1 (Please, see the revised version) alongside the IF1/complex V ratio. In addition, we have now performed WBs in the BAT exposed for 3 and 7 days to thermoneutrality (~28°C) where we found that IF1 is not reduced following whitening of BAT by this approach whilst UCP1 and other mitochondrial proteins are reduced. This set of data is now included in Figure 1I,K,L.

2. In Figure 2J, the drop in MMP is lower upon adrenergic stimulation than in Figure 2E. The same observation applies to other results when the reduction in MMP after NE addition is minimal. Why do the authors remove TMRM for the measurements of membrane potential? TMRM imaging is normally done in the presence of the dye in non-quenching mode. Treatments should be done prior to the addition of the dye and then TMRM should be added and left during the imaging analysis and measure in non-quenching mode. This might explain some of the above-mentioned points regarding the MMP data. Alternatively, if the dye is removed before the measurements, they should let the cells to adapt and so the dye equilibrates between mitochondria and cytosol. A more elegant method to measure membrane potential could be live-cell imaging. In addition, authors propose that mitochondrial membrane potential upon NE stimulation is maintained by reversal of ATP synthase. If this is the case, one would expect that addition of oligomycin in NE treated adipocytes would cause depolarization. However, in FigS2A this is not the case. Authors should comment on this in addition to considering more elegant approach to measure MMP.

RESPONSE: We apologize for the lack of details in the methods. All treatments (i.e., transfection and norepinephrine stimulation) were performed before the addition of TMRM. Indeed, this approach does not have the resolution compared to safranin in isolated mitochondria (Fig. 1D), which limits our interpretation regarding the dynamic role of IF1 on MMP in brown adipocytes. We have taken care to state the limitations of our method throughout the entire paper to avoid overinterpretation of our data. Regarding the removal of the dye before the measurements, our internal controls indicate that this procedure does not change the ability of our method to detect fluctuations in MMP (i.e., oligomycin and FCCP as internal controls). Nevertheless, as suggested by the reviewer, to test the time effect of the probe equilibrium (i.e., mitochondria versus cytosol) in our method, we loaded cells with TMRM 20 nM for 30 min and measured the fluorescence right after the removal of the probe/washing steps for another 10 min. We were not able to detect differences in the fluorescence in a time-dependent manner (see below). Therefore, we conclude the removal of TMRM does not influence the fluorescence of the probe in differentiated brown adipocytes. In addition, we performed a similar experiment using TMRM in the quenching mode (200 nM),

however, after the removal of TMRM, we added FCCP (1 μ M) to the cells for 10 min under constant agitations at 37°C. This approach aimed to expel all TMRM that accumulated within the mitochondria in an MMP-dependent manner. Therefore, excluding the dynamic Brownian movement that we could have caused by the removal of the dye before the measurement mentioned by the reviewer. By doing this, we found the same effect of IF1 overexpression in the reduction of MMP in the presence of norepinephrine.

Protocol:

- Transfection (24h) on day 4 of differentiation + 24h just normal media
- 30 min norepinephrine 10 μ M
- 200 nM TMRM on top of NE
- Washing step
- Add FCCP 1 μ M for 10 min, and read (The aim here was to release all TMRM accumulated inside of mitochondria in a MMP-dependent manner)

In summary, the data suggests the removal of the dye from the cells does not influence the fluorescence of TMRM, therefore, enabling us to make conclusions regarding the biological effects of IF1 manipulation in the MMP of brown adipocytes. Regarding the reverse mode of ATP synthase and the absence of effects with oligomycin, given oligomycin inhibits both rotation of ATP synthase and even uncoupled brown adipocytes respond to oligomycin (i.e. reduction in O₂ consumption), the prediction of lowering MMP in the presence of oligomycin due to inhibition of the reserve mode of ATP synthase is more complicated than anticipated. Nevertheless, we have now addressed this topic in the discussion section. Lastly, we generally observe a reduction in MMP around 10-25% in differentiated adipocytes upon NE treatment (30 minutes, 10 μ M). However, due to the differentiation state of the cells, MMP response from norepinephrine fluctuated from experiment to experiment. Therefore, we did not compare experiments performed on different days or batches, but only within the same differentiation batch to reduce variability.

3. In Figure 2, in the model of siIF1, there is baseline more phosphorylation of AMPK than in the scramble control (pAMPK). However, this is not the case of p-p38MAPK. Do the authors have any explanation for those differences in baseline activation of the stress kinases when IF1 is silenced? In the same experimental group, addition of NE seems to have more effect in the scrambled than in siIF1, but the plotted data does not reflect these differences. In contrast, increase in pAMPK upon NE is higher in IF1 overexpressing cells compared to EV (Figure 2H), but again this is not reflected in western blot quantification (Figure 2I).

RESPONSE: Although some differences in pAMPK in the treatments were observed as gathered by the representative blots, these changes were not confirmed later in different biological replicates, therefore, the overall effect of IF1 manipulation in pAMPK does not change. Given we used this approach as quality control for our experiments to guarantee norepinephrine treatment works, we removed the pAMPK data from the study and kept p38 as a marker of adrenergic signaling activation (please see revised Fig. 2 in the main file).

4. Does NE promote decrease of IF1 expression in control (siScramble and EV) adipocytes? The authors should test it and see whether it goes in the same direction as the observations derived from the experiments in cold exposed mice. This is very important point, as it could explain the lack of an additional effect of IF1 silencing on NE-induced depolarization (Figure 2E).

RESPONSE: We thank the reviewer for this suggestion. In line, with the *in vivo* data, acute NE treatment in differentiated brown adipocytes does not change IF1 mRNA and protein levels. We have now added this information and the corresponding interpretation to the updated manuscript.

5. Does NE promote decrease of IF1 expression in the scramble and EV adipocytes? The authors should test it and see whether it goes in the same direction as the observations derived from the experiments in cold exposed mice.

RESPONSE: As this question is the same as #4, we believe the reviewer may have erroneously pasted this here.

6. For MMP data in Fig2, they should include significance between non treated and NE-treated groups. They say: "While UCP1 ablation did not cause any effect on MMP upon adrenergic stimulation...", but NE caused (probably significant) depolarization in siUCP1, which seems even stronger than depolarization in EV. This is opposite to what you would expect. They also didn't confirm UCP1 silencing with western blot.

RESPONSE: We thank the reviewer for this suggestion. We have now included the expected statistical main effect of NE upon MMP. Although the effects of IF1 overexpression were blunted when Ucp1 was silenced, we indeed still observed the same degree of reduction in MMP in brown adipocytes. This finding has two possible explanations, one is the effectiveness of the silencing protocol, therefore, residual Ucp1 expression may still play a role in this experiment; second, other ATP-consuming processes are able to lower MMP in a UCP1-independent manner. We have added this information to the updated manuscript to make the reader aware of our findings as well as the limitations of the method. Unfortunately, we were not able to detect UCP1 protein levels due to technical issues. Given the effects of IF1 overexpression were blunted when Ucp1 was silenced, we believe this functional outcome is sufficient, alongside mRNA levels, to demonstrate the effectiveness of our silencing protocol.

7. It has been established that decreased expression of IF1 promotes increase in the reverse activity of Complex V, ATP hydrolytic activity. Increase in ATP hydrolysis also affects ECAR. The authors should consider this when calculating the contribution of ATP glycolysis versus ATP OXPHOS since the ATP hydrolysis is also playing a role in the ECAR increase. The data should be reinterpreted. ATP hydrolysis should be measured in the situation where IF1 is silenced and overexpressed. These measurements can be done in cells using the seahorse.

RESPONSE: The only differences we observed in MMP are in the presence of norepinephrine (i.e. UCP-1-dependent proton conductance), which is not present during the estimation of ATP production by Seahorse analysis. Nevertheless, we have now improved the description of our experimental protocol and limitations to estimate ATP production to make it as clear as possible to the reader. Lastly, given the addition of *in vivo* gain-of-function experiments, we have now determined the ATP hydrolytic activity in this model, which offers a better understanding of the *in vivo* modulation of IF1 levels affecting ATP synthase activity (reverse mode). We hope the reviewer understands our motivation to focus on the *in vivo* model of gain-of-function regarding ATP synthase activity.

8. The authors use GAPDH as loading control in western blots. They should use another protein since GAPDH is part of the intermediary metabolism and plays a role in glycolysis.

RESPONSE: We understand the concern of the reviewer regarding the use of GAPDH as a loading control for the studies of metabolism. However, as can be observed by the western blot images, GAPDH levels do not change in our experimental models, therefore, we feel confident that our loading is homogeneous throughout our gels.

9. The authors show that reduction of IF1 involves more lipid utilization. They should include more experiments showing the connection of the metabolic adaptation in the absence of IF1 and some lipid imaging.

RESPONSE: We appreciate this suggestion. We have now performed Oil Red O staining in differentiated adipocytes following ablation of IF1. However, we did not observe any effect on lipid accumulation in primary brown adipocytes following IF1 knockdown. Therefore, the effects of IF1 ablation on lipid mobilization are not due to lipid content or reflected in lipid accumulation. We have now added this new information to the manuscript (please, see the revised form Fig. EV3).

10. In the text, "Despite this adjustment of experimental conditions, we did not detect any effect of IF1 ablation on mitochondrial oxygen consumption (Supplementary Fig. 3A,B)", this is true for baseline, NE-driven and ATP-linked respiration, but what about maximal respiration? There is a huge increase in IF1 knockdown... They should explain these results.

RESPONSE: We perform this experiment to address the question of whether the lipid mobilization induced by norepinephrine would uncouple mitochondria in a UCP1-independent manner. Given the absence of effect between scrambled and IF1 ablated cells in mitochondrial respiration in the presence of norepinephrine and following the addition of oligomycin, we concluded no effect of lipolysis-induced UCP1-independent uncoupling. However, as observed by the reviewer and consistent with other data within the study, the interaction between lipid metabolism and IF1 knockdown seems to affect maximal electron transport chain activity, which although interesting, was not the focus of the present study. Nevertheless, we have now acknowledged these findings and a possible explanation for them in the revised manuscript.

11. In Figure 3K they present OCR as % of baseline, but in a similar experiment in Figure 4G it is OCR/protein, they should make the Y axis consistent across experiments.

RESPONSE: We apologize for this overlook. We have now edited all the axes and labels for consistency.

12. The graphical abstract is confusing. In BAT there are two populations of mitochondria, the cytosolic and the mitochondria attached to the lipid droplet, peridroplet mitochondria (PDM). Upon adrenergic stimulation, PDM leave the lipid droplet and lipolysis takes place. The authors propose that upon adrenergic stimulation, IF1 is reduced and there is lipid mobilization. The part of the scheme where it says "fully recruited" should be removed or rewritten, since adrenergic stimulation is not compatible with mitochondria recruitment around the lipid droplet.

RESPONSE: Thank you for this input. Given the addition of new experiments and interpretation, we have now redrawn the graphical abstract and addressed this topic in the discussion section.

13. The title should be rewritten to better reflect the research presented in the manuscript.

RESPONSE: Thank you for this input. Given the addition of new experiments, we have now rewritten the title accordingly.

Minor comments:

14. Some of the Y axis should be corrected. For example, in Figure 2J, L and M should say % of EV untreated, Similarly, in Figure 2E, it should say % of scramble untreated. In Figure 3N, the Y axis is misspelled. All the Y axis referring to percentages should have the same scale for comparison purposes.

RESPONSE: Thank you for the proofreading. We have now edited the scales and labels to keep consistency.

15. The authors should describe better the results corresponding to Figure 2. There is a lot of information and they should improve the description pertaining the connection between the different pieces of data relating the different signaling pathways that are shown. For westerns

in this Figure, they should provide some rationale (one to two sentences in the results section) as to why they are checking the expression of pAMPK and p38-MAPK.

RESPONSE: We have now edited the description of our results to make them as clear as possible.

Here are some comments referring to the methods section:

16. For Complex V hydrolytic activity, the reaction buffer contains 10mM Na-azide. I guess this is to inhibit respiration, but wouldn't azide also inhibit complex V at this concentration?

RESPONSE: We thank the reviewer for this question. To test that, we performed complex V activity in buffers containing or not 10 mM sodium azide. As demonstrated below, the presence of sodium azide in the buffer does not influence complex V activity in two different tissues with low and high complex V activity (BAT and heart, respectively).

Table 1. ATP synthase hydrolytic activity in the presence or absence of Na-azide.

	BAT	Heart
+Na-azide	100 ± 43.01	100 ± 39.36
-Na-azide	82.6 ± 4.33	111.3 ± 43.32
+Na-azide + oligomycin	15.3 ± 4.32*	13.8 ± 14.01*
-Na-azide + oligomycin	14.2 ± 3.53*	11.9 ± 2.88*

Data presented as % of control (i.e. presence of Na-azide and absence of oligomycin) for both tissues independently. N = 2-3/condition. Statistical test: two-way ANOVA. * main effect of oligomycin ($p < 0.05$).

17. In the mitochondrial isolation protocol, they say "mitochondria were centrifuged at 800g for 10min..." Will this speed pellet the mitochondria? I think this is a mistake in writing.

RESPONSE: We apologize for the lack of clarity. What was centrifuged at 800 g was the whole-tissue homogenate to discard cellular debris, before pelleting mitochondria at 5000 g. We have now corrected this mistake in the methods section.

18. For the safranin-O experiment, they don't mention mitochondrial substrate used, probably it's in the reference that they provide, but I think it should be included in the text.

RESPONSE: We did not use any substrate because our goal was to test the contribution of ATP synthase to mitochondrial membrane potential. For that, we inhibited proton movement within the ETC with antimycin A and through UCP1 with GDP (see Methods). We have now edited our Method's description to make sure the reader is aware of our approach.

Reviewer #3 (Significance (Required)):

The manuscript is well written, and it flows well when reading. However, there are some additional experiments that need to be performed to reach the conclusions the authors claim.

RESPONSE: We thank the reviewer for the positive commentaries regarding our work and hope to have answered the open questions with the edits and new experiments.

The role of ATP hydrolysis in BAT thermogenesis is novel and interesting as it can shed some light onto potential approaches to promote BAT activation.

Reviewer #4 (Evidence, reproducibility and clarity (Required)):

This is an interesting investigation into the activity of IF1 in brown adipocytes. The findings are innovative and the conclusion is well-supported by the data. The conclusion is in line with previous reports on IF1 activities in other cell types, particularly in terms of its regulation of FoF1-ATPase. The authors have executed an exceptional job in designing the study, preparing the figures, and writing the manuscript. Overall, this study significantly contributes to the understanding of IF1 activity in brown adipocytes and its role in thermogenesis.

RESPONSE: We thank the reviewer for the kind words. Please, find below our answers in a point-by-point manner.

Reviewer #4 (Significance (Required)):

The study demonstrates involvement of IF1 in regulating thermogenesis in brown adipocytes, which is a unique aspect not covered in existing literature. Advantage of the study is well-designed cellular studies. The major weakness is lack of proof of conclusion *in vivo*. There are a few minor concerns that should be addressed to further enhance quality of the manuscript.

RESPONSE: We have now included two *in vivo* models, whole-body IF1 KO mice and BAT-injected IF1 overexpression to test the role of IF1 in BAT biology. The whole dataset is included in the main manuscript, where we conclude the BAT IF1 overexpression partially suppresses β 3-adrenergic induction of thermogenesis alongside a reduction (overall and UCP1 dependent) in mitochondrial oxygen consumption. Also, similar to our *in vitro* experiments, IF1 KO mice did not present any difference in adrenergic-stimulated oxygen consumption.

1. Current discussion does not mention the regulation of IF1 protein by the cAMP/PKA pathway. This point should be included to provide a comprehensive understanding of the regulatory mechanisms of IF1 protein.

RESPONSE: Thank you for this suggestion. We have now added this topic to the discussion.

2. It has been reported that IF1 also influences the structure of mitochondrial crista. Considering the observed changes with IF1 knockdown, it would be valuable to discuss this activity in relation to the findings of the study.

RESPONSE: We discussed the implications of IF1 modulation in mitochondrial morphology in the revised manuscript.

Dear Dr. Alexander Bartelt,

Thank you for submitting your revised manuscript (EMBOJ-2024-117748-T | [RC-2023-02105] R) to The EMBO Journal. As mentioned, your amended study was sent back to the referees for their re-evaluation, and we have received comments from two of them, which I enclose below. Please note that while referees #3 and #4 were unfortunately not able to re-evaluate the work at this time, we have editorially assessed your response to their concerns raised and found them to be addressed satisfactorily. As you will see, the other experts stated that the work has been substantially improved by the revisions and they are now in favour of publication, pending minor revision.

Thus, we are pleased to inform you that your manuscript has been accepted in principle for publication in The EMBO Journal.

Please consider the remaining technical point by referee #2 carefully, amending the study with additional experiments and adjusting the text where appropriate.

Also, we now need you to take care of a number of minor issues related to formatting and data presentation as detailed below, which should be addressed at re-submission. I will share these within the few next days.

Please contact me at any time if you have additional questions related to below points.

Thank you for giving us the chance to consider your manuscript for The EMBO Journal. I look forward to your final revision.

Again, please contact me at any time if you need any help or have further questions.

Best regards,
Daniel Klimmeck

Referee #1:

The major claims made are now well supported by the evidence. The manuscript is very well written, figures are clear and succinct. Overall, the quality of the work is very high. Given that IF1 is implicated across many fields of study, the novel discovery of IF1 as a regulator of brown adipose mitochondrial bioenergetics will be of significance across several fields. Having reviewed the initial submission, the authors have done an excellent job of addressing all concerns raised with the initial submission.

Referee #2:

Summary, claims and conclusions:

The revised version of the manuscript by Brunetta and colleagues has been improved attending to the requests and suggestions of the reviewers to the experiments carried out in cultured brown adipocytes. Moreover, incorporated new data of AVV-mediated IF1 loss and gain-of-function experiments in BAT of mice. Overall, the findings support that the mitochondrial dose of IF1 is implicated in controlling bioenergetics in brown adipocytes, which is in line with previous findings in other IF1 loss and gain-of-function experiments in mitochondria of different mouse tissues and cells. Specifically, in brown mitochondria it is suggested that the loss of IF1 expression upon cold adaptation facilitates the reversal of ATP synthase (hydrolytic mode) to contribute to the maintenance of the mitochondrial membrane potential, which is the driving force that supports UCP1-mediated non-shivering thermogenesis.

Two major arguments were pointed out in my previous review of this work. They were related with the assessment of the relative expression level of IF1 in mouse tissues (liver, heart and BAT), and with the differential migration and labeling of IF1 in polyacrylamide gels (12, 15, 17KDa bands). These are key points to accept what is being communicated in Fig. 1 of the paper. I am not questioning the data of IF1 loss and gain of function experiments using brown adipocytes, which seem to be devoid of IF1 unless the overexpression of the protein is triggered (see EV in Fig. 2G, 4B). However, in response to my requests, the authors indicated that the heterogeneity of IF1 bands resulted from the differential migration of IF1 in custom-made (Fig. 1 & 5) and commercially available pre-cast (Fig. 2,3, 4) polyacrylamide (no agarose) gels. If that is the situation, then it is mandatory to show a blot using pre-cast gels documenting the expression of IF1 as a 12 KDa band in liver, heart and BAT. With this blot, readers will verify the differential expression of IF1 in mouse tissues with confidence, excluding the possibility that extracts from different mouse tissues could express non-specific 17 KDa immunoreactive proteins towards the antibody used, mistakenly ascribed to IF1 immunoreactivity.

Rev_Com_number: RC-2023-02105

New_manu_number: EMBOJ-2024-117748-T

Corr_author: Bartelt

Title: IF1 is a cold-regulated switch of ATP synthase to support thermogenesis in brown fat

Dear Alexander,

Further to below message please find enclosed the formatting changes required for your final manuscript version.

Please let me know any time should there be additional questions related to this.

Best regards,
Daniel

>> Author Contributions: Please remove the author contributions information from the manuscript text. Note that CRediT has replaced the traditional author contributions section as of now because it offers a systematic machine-readable author contributions format that allows for more effective research assessment. and use the free text boxes beneath each contributing author's name to add specific details on the author's contribution.

More information is available in our guide to authors.
<https://www.embopress.org/page/journal/14602075/authorguide>

>> Adjust the title of the 'Competing Interest' section to 'Disclosure and Competing Interests Statement'.

>> Authors: revisit discrepancy between Pedro M.M. Moraes-Vieira (manuscript) and Pedro Vieira (eJP): please confirm which is correct.

>> Section order should be corrected as follows: title page with complete author information, abstract, keywords, introduction, results, discussion, methods, data availability section, acknowledgements, disclosure and competing interests statement, references, main figure legends, tables, expanded figure legends.

>> Please provide the main manuscript text as .docx file.

>> The list of abbreviations should be removed from the manuscript. Please introduce abbreviations in the running text at first appearance.

>> Figures in separate files: Figures should be removed from the manuscript and uploaded as individual, high resolution figure files. The legends should be compiled at the end of the manuscript text, after the References. The six EV figures should also be removed from the manuscript and uploaded as individual files. Their legends should be moved after the main figure legends, with the heading Expanded View Figure Legends.

>> Data availability section: please enter a Data availability section into the manuscript, stating 'No large-scale data amenable to data repository deposition were generated in this study.' .

>> Source data: provide a complete set of source data for the study according to separate instructions by my colleague Hannah Sonntag. Supplementary Fig. 1 is source data and should be removed from the manuscript.

- > The sentence "The graphical abstract and cartoons in Figures 1 and 5 were created using BioRender.com." should be removed from the Acknowledgements and instead added to the Methods in a separate "Graphics" section.
- > Reference list format: please correct to 10 author names listed before et al. . DOIs should be removed for all published articles.
- > Appendix file: The supplementary tables should be removed from the manuscript and uploaded as one Appendix file in PDF format. They should be renamed 'Appendix Table S1' and 'Appendix Table S2'. The appendix file will need a table of contents (with page numbers) to be added to the first page.
- > Funding: information on funding is incomplete in our online system. All funders, including project numbers, need to be entered into our system in addition to being mentioned in the Acknowledgements.
- > Please remove the graphical synopsis from the manuscript and submit it as separate image file.
- > Synopsis: please provide 2-5 one-short-sentence bullet points that summarise the article.
- > Author checklist: provide a completed author checklist for your study.
- > Consider additional changes and comments from our production team as indicated below:

- Data Availability Section

Please note that the data availability statement is not provided in the manuscript.

- Figure legends:

1. Please note that the exact p values are not provided in the legends of figures 1b, e, g-h, j-k; 2a, f, j-m; 3a, f, i-j, l, n-o; 4a, d, f, h, j-k; 5b, e, g, i, l-m, p-r; EV 1a-b; EV 2a-b; EV 4b; EV 6c, e, h-i.
2. Please note that for the figures EV 3a-b; EV 5a-b, d-f, h; p-values and statistical tests are indicated in the legends. However, comparison for the same, "" has not been represented in the figures. Please rectify this in the figures or legends as applicable.

The authors addressed the remaining editorial issues.

Dear Alexander,

Thank you for submitting the revised version of your manuscript. I have now evaluated your amended manuscript and concluded that the remaining minor concerns have been sufficiently addressed.

I am thus pleased to inform you that your manuscript has been accepted for publication in the EMBO Journal.

Related, I kindly ask for your consent on keeping the referee figure included in this file.

On a different note, I would like to alert you that EMBO Press offers a format for a video-synopsis of work published with us, which essentially is a short, author-generated film explaining the core findings in hand drawings, and, as we believe, can be very useful to increase visibility of the work. Please see the following link for representative examples and their integration into the article web page:

<https://www.embopress.org/doi/full/10.15252/emj.2019103932>

Finally, we have noted that the submitted version of your article is also posted on the preprint platform bioRxiv. We would appreciate if you could alert bioRxiv on the acceptance of this manuscript at The EMBO Journal in order to allow for an update of the entry status. Thank you in advance!

Best regards,

Daniel

Daniel Klimmeck, PhD
Senior Editor
The EMBO Journal
EMBO
Postfach 1022-40
Meyerhofstrasse 1

D-69117 Heidelberg
contact@embojournal.org
Submit at: <http://emboj.msubmit.net>
